# A comprehensive suite for extracting neuron signals across multiple sessions in one-photon calcium imaging

Pablo Vergara [1,7] ✉, Yuteng Wang[1,2,7], Sakthivel Srinivasan[1], Zhe Dong[3], Yu Feng[3], Iyo Koyanagi[1], Deependra Kumar[1], Yoan Chérasse [1], Toshie Naoi[1], Yuki Sugaya[4,5], Takeshi Sakurai [1], Masanobu Kano [4,5], Tristan Shuman[3], Denise Cai [3], Masashi Yanagisawa [1,2,6] & Masanori Sakaguchi [1,2,6] ✉

We developed CaliAli, a comprehensive suite designed to extract neuronal signals from one-photon calcium imaging data collected across multiple sessions in free-moving conditions in mice. CaliAli incorporates information from blood vessels and neurons to correct inter-session misalignments, making it robust against non-rigid brain deformations even after substantial changes in the field of view across sessions. This also makes CaliAli robust against high neuron overlap and changes in active neuron population across sessions. CaliAli performs computationally efficient signal extraction from concatenated video sessions that enhances the detectability of weak calcium signals. Notably, CaliAli enhanced the spatial coding accuracy of extracted hippocampal CA1 neuron activity across sessions. An optogenetic tagging experiment showed that CaliAli enhanced neuronal trackability in the dentate gyrus across a time scale of weeks. Finally, dentate gyrus neurons tracked using CaliAli exhibited stable population activity for 99 days. Overall, CaliAli advances our capacity to understand the activity dynamics of neuronal ensembles over time, which is crucial for deciphering the complex neuronal substrates of natural animal behaviors.

Miniaturized microscopes have revolutionized the ability to observe neuronal dynamics in freely behaving animals over long periods of time[1]. Long-term studies in which neuron populations are tracked across separate recording sessions require dedicated algorithms to identify neurons accounting for non-rigid brain deformations and fluctuations in neuron activity between sessions. Typically, neuronal tracking algorithms in one-photon calcium (Ca²⁺) imaging address this issue by aligning neuronal footprints (i.e., visual representations of neuron shape and location in the field of view (FOV)) obtained independently during distinct recording sessions[2,3]. These algorithms subsequently employ probabilistic models to identify the same neurons across sessions based on their spatial and temporal properties.

One drawback of these algorithms is that they rely on independent neuronal extraction in each recording session. In one-photon Ca²⁺ imaging, a neuron's detection largely hinges on both its activity level and background signal fluctuations. Consequently, neurons that are

[1]International Institute for Integrative Sleep Medicine (WPI-IIIS), University of Tsukuba, Tsukuba, Ibaraki, Japan. [2]Doctoral Program in Neuroscience, Degree Programs in Comprehensive Human Sciences, Graduate School of Comprehensive Human Sciences, University of Tsukuba, Tsukuba, Ibaraki, Japan. [3]Nash Family Department of Neuroscience, Icahn School of Medicine at Mount Sinai, New York, USA. [4]Department of Neurophysiology, Graduate School of Medicine, The University of Tokyo, Tokyo 113-0033, Japan. [5]International Research Center for Neurointelligence (WPI-IRCN), The University of Tokyo Institutes for Advanced Study (UTIAS), Tokyo 113-0033, Japan. [6]Faculty of Medicine, University of Tsukuba, Tsukuba, Ibaraki, Japan. [7]These authors contributed equally: Pablo Vergara, Yuteng Wang. ✉e-mail: pablo.vergara.g@ug.uchile.cl; sakaguchi.masa.fp@alumni.tsukuba.ac.jp

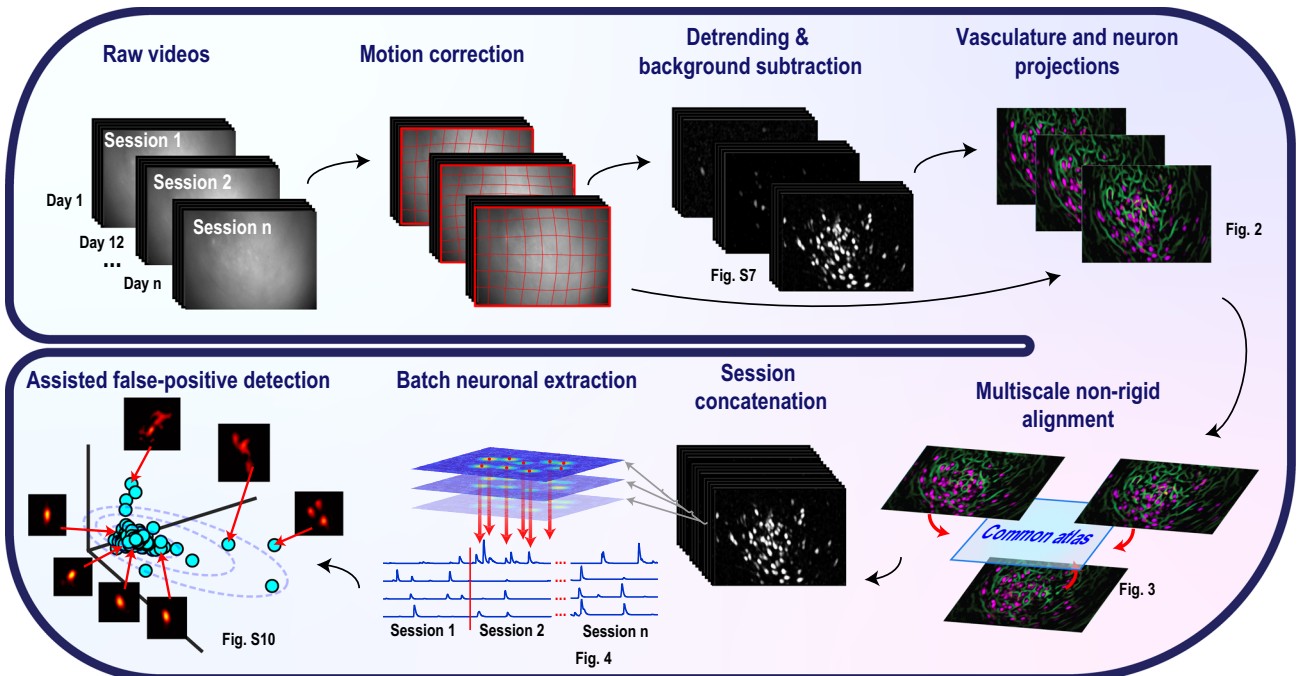

**Fig. 1 | CaliAli: A complete computational suite for processing long-term Ca²⁺ imaging data.** Each raw Ca²⁺ imaging video is subjected to motion correction, detrending, and noise scaling—steps tailored for downstream inter-session alignment and subsequent extraction from concatenated video sequences. CaliAli then extracts projections of neurons and BVs from each processed session. These projections are then used in a multi-level, group-wise registration process that aligns each session to a common atlas via the appropriate non-rigid displacement field. Subsequently, all videos are concatenated, and an optimized non-negative matrix factorization module extracts neuronal signals from this combined video stack. This memory-optimized module is robust against neuronal signal dilution common in processing large video stacks. Finally, an additional module automatically sorts neurons based on shape quality, enhancing the identification of false-positive detections.

successfully identified in one session may not be detected in subsequent sessions. Although misalignment due to affine transformations (e.g., translations and rotations) can be rectified with a minimal number of matching neuronal footprints, non-rigid transformations necessitate a larger number of stable features distributed across the entire FOV[4]. Indeed, a substantial portion of inter-session misalignments are non-rigid (Supplementary Figs. 1a–c), which increases with longer temporal gaps between sessions (Supplementary Fig. 1d). This issue is worsened by the fact that neurons have homogenous blob-like shapes that can easily lead algorithms to converge into local minima. That is, different neurons can appear highly correlated due to their uniform shapes, leading them to be mistakenly recognized as the same neuron.

Beyond these challenges, mismatched neuronal footprints between sessions hinder data interpretability. As it is often impossible to distinguish between neurons that are inactive and those that are simply undetected, most studies restrict their post-extraction analysis to neurons that can be consistently identified across all sessions[3,5]. However, such approaches could overlook relevant activity patterns that occur during brain processes in which neuron activity naturally changes across sessions, such as contextual remapping[6], representational drift[7], and memory consolidation[8].

An alternative, although less common, approach to neuronal tracking across sessions entails concatenating sessions, applying motion correction algorithms to rectify deformations within and across sessions, and extracting neuronal signals from the concatenated video[9–12]. However, video concatenation often leads to poor signal extraction[3]. For example, although motion correction algorithms can mitigate motion artifacts within individual sessions, they fall short in addressing inter-session misalignments. Such misalignments are particularly challenging due to variations in the FOV between sessions, including changes in baseline fluorescence and the z-axis. Furthermore, even with perfect video alignment, changes in

baseline fluorescence can introduce abrupt changes in pixel intensity at the junction of two sessions, which can compromise subsequent neuronal extraction.

An additional challenge in video concatenation is signal dilution, as low signal-to-noise ratio (SNR) transients from neurons that are highly active in one session but less active in other sessions may be masked or diminished. This occurs because the extraction of neuronal signals relies on the relative contribution of a neuron's activity to overall fluctuations in the FOV. Therefore, concatenation can obscure the detectability of low-SNR transients that would otherwise be efficiently isolated and analyzed if the sessions were processed independently. This necessitates novel approaches to ensure that significant neuronal activity is not overshadowed by concatenation. Yet another drawback of concatenation lies in its elevated computational demands, as many video frames must be processed together. This contrasts with footprint alignment methods, which process each session independently and require fewer computations.

To address these issues, we developed CaliAli (Calcium Imaging intersession Alignment), a comprehensive suite designed for neuronal tracking across multiple sessions (Fig. 1). CaliAli implements a concatenation-based approach that leverages information from both blood vessels (BVs) and neurons for inter-session alignment and integrates an extraction pipeline specialized for concatenated videos, guaranteeing resilience against signal dilution. CaliAli operates seamlessly in batch mode, effectively overcoming computational constraints while upholding the quality of signal extraction. Additionally, it features a straightforward yet efficient post-processing interface for identifying false-positive components.

## Results

CaliAli repurposes BV segmentation techniques used in ocular imaging[13] to accentuate intrinsic BVs in one-photon Ca²⁺ imaging

datasets. CaliAli integrates these BV images with projections of active neurons from each session to compute displacement fields for aligning videos across multiple sessions. The aligned videos are processed using a computationally efficient approach to extract long-term tracked Ca²⁺ signals.

## CaliAli uses diffeomorphic demons to calculate displacement fields

CaliAli utilizes diffeomorphic demons[14] to correct inter-session misalignment (Fig. 1, multiscale non-rigid alignment). The Demons algorithm, in its basic form, iteratively warps one image toward another while minimizing a metric that quantifies the dissimilarity between images. Diffeomorphic demons[15] expands on this operation by forcing the deformation field it generates to remain diffeomorphic (i.e., smooth and continuous) throughout the alignment process.

## Enhancement of BV structures through Hessian filters

In practice, implementing diffeomorphic demons to correct inter-session misalignments involves creating a representative image from each motion-corrected session (i.e., video projection). This representative image is used to calculate a displacement field applied to each frame in the session. Because neurons are the ultimate structure for alignment, a plausible approach would be to use a projection reflecting active neurons in the FOV. However, alignment algorithms can easily converge on local minima when relying solely on neuron shapes for registration. One potential landmark structure prevalent in Ca²⁺ data that could overcome these limitations is BVs. BVs are intimately linked to neuron locations[16] but exhibit intricate and diverse shapes that favor accurate alignment. Although some motion correction algorithms indirectly benefit from BVs by using Gaussian filters to enhance high-contrast areas within the FOV[17], these methods lack specificity for BVs and may incorporate other neuropile signals without the same inter-session stability. Therefore, CaliAli utilizes Hessian filters[13] that selectively enhance dark, elongated structures within the FOV, thereby maximizing the detection of faint BVs (Fig. 1, vasculature and neuron projections). This allows for creation of a representative projection of BV structures observed in each session.

## Improved inter-session alignment with a multiscale approach combining BVs and neurons

CaliAli utilizes a multilevel alignment approach[18] incorporating BVs and neurons to correct inter-session misalignments (Fig. 2a). This method entails the creation of an image pyramid from the original data. Each level of the pyramid consists of a down-sampled version of BVs derived using Hessian filters and neuron projections. These neuron projections are obtained by combining two images: the correlation image and the peak-to-noise ratio (PNR) image (i.e., maximum projection of a noise-scaled video). The correlation image, calculated as the correlation between adjacent pixels across time, enhances the spatial structure of active neurons because pixels within a neuron tend to be more correlated with each other than with surrounding neuropil or inactive areas. This approach to enhancing neuron shapes is commonly used in Ca²⁺ imaging analysis and has previously been employed in methods such as Constrained Nonnegative Matrix Factorization (CNMF)-E[19]. The PNR image highlights pixels with high temporal variance relative to noise, further aiding neuron detection.

As we descend the pyramid, image resolution increases, resulting in larger images at each tier. The upper levels entail lower-resolution images of BVs and/or neurons, whereas the lower levels entail higher-resolution images of only neurons. This multilevel alignment strategy adheres to a coarse-to-fine approach, commencing with the alignment of BVs at the coarsest level (Fig. 2a, top of the pyramid), which, due to their lower resolution, are more resilient against initial misalignments and noise. Following coarse alignment, transformation parameters undergo refinement and serve as initial estimations for the next higher-

resolution level of the pyramid. This continues iteratively until achieving final alignment at the highest resolution level, at which images closely match in scale and intricate details (Supplementary Video 1).

Note that this final step primarily relies on neuron projections after most misalignments are estimated using BVs. If only neurons were utilized at each level, there would be a risk of convergence toward the nearest neighbor instead of the genuine neuronal pair (Fig. 2b). To assess the effectiveness of BVs in preventing this scenario, we created simulated Ca²⁺ imaging sessions that faithfully mirror the temporal and spatial attributes of actual experiments (Supplementary Fig. 2; Supplementary Video 2) and for which we have exact knowledge of neuron locations and activities (i.e., simulated ground truth). To incorporate actual BV structures, we used frames obtained from Ca²⁺ imaging experiments as the constant baseline image of each session (Supplementary Video 3). We simulated two purposely misaligned sessions to assess our multilevel alignment method, applying both BV and neuron projections (CaliAli) or only neuron projections (Fig. 2c). The use of both BV and neuron projections resulted in greater reductions in the centroid distance between true neuron pairs compared with the use of only neuron projections. This highlights that integrating BVs effectively mitigates entrapment within local minima during the iterative refinement of non-rigid displacement.

To evaluate our multiscale registration strategy, we compared the post-alignment stability of neuron projections across seven imaging sessions spanning 99 days (Figs. 2d; Supplementary Fig. 3a, b). We tested various registration algorithms commonly used in one- and two-photon analysis packages[2,3,17,20–24] (Fig. 2e–f). Algorithms not specifically designed for one-photon microscopy were adapted by applying a spatial filter to the mean frame[17,23] or using the neuron/BV projections from CaliAli. Alignment stability was evaluated by two metrics: the average spatial correlation of aligned projections and the average sharpness of the mean aligned projection ("crispness"), as previously described[17,20,23,25]. CaliAli's alignment strategy outperformed other methods on both metrics (Fig. 2e). Furthermore, we confirmed that this improvement is specifically attributable to the combined use of BV and neuron projections. Using other projections or BV and neurons independently with CaliAli's alignment strategy yielded lower performance with simulated imaging data (Supplementary Fig. 3c-f).

## Changes in BVs over time and their influence on neuronal tracking

Variations in BV structures across time may compromise alignment accuracy. We estimated BV structural similarity using a score based on the standard deviations between spatial correlations of aligned BV projections and those of BVs obtained through random non-rigid misalignment within the FOV (Fig. 3a). To examine changes in the BV similarity score over time, we imaged the hippocampal dentate gyrus (DG) using two approaches: 1) maintaining a constant focal plane across sessions or 2) adjusting the focal plane for each session based on visual inspection of the FOV. The BV similarity score declined steadily for both methods; however, dynamic adjustment of the focal plane slowed this decline (Fig. 3a). These results suggest that a substantial portion of variations observed in BVs over time may be attributed to a FOV shift along the z-axis.

To assess an appropriate threshold value for the BV similarity score for inter-session alignment, we simulated Ca²⁺ imaging sessions in which we systematically changed the BV correlation across sessions. To achieve this, the baseline image from one session was systematically merged with a baseline image from a different FOV (Fig. 3b). To evaluate CaliAli's performance, we compared the temporal similarity of the simulated ground truth Ca²⁺ traces with those derived from CaliAli (Fig. 3c; Supplementary Fig. 4). Given that the reconstruction of temporal traces is never flawless, we judged Ca²⁺ traces with a cosine similarity to the simulated ground truth > 0.8 as true-positive reconstructions. We did not find meaningful changes in

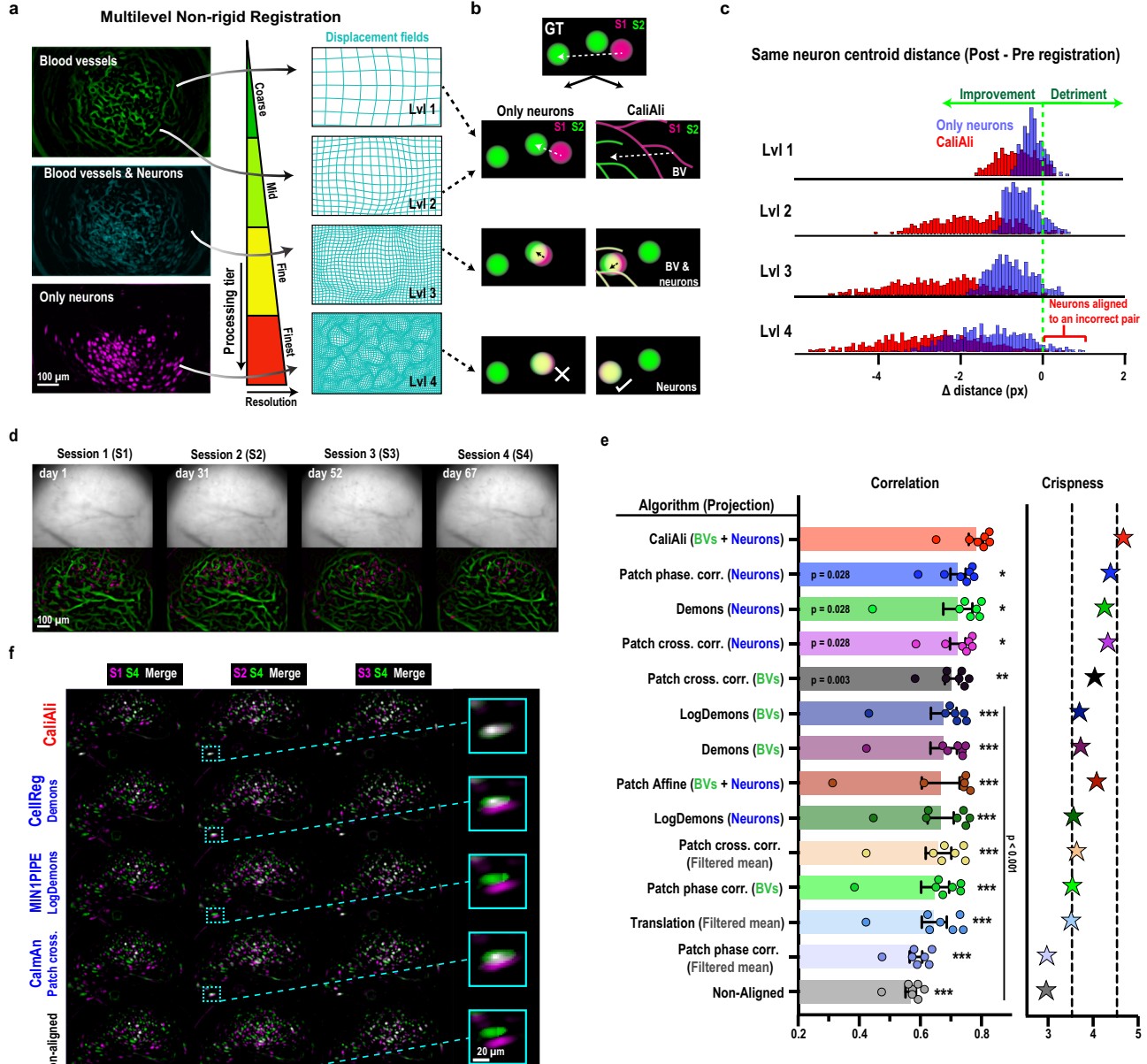

**Fig. 2 | CaliAli enhances inter-session registration through a multiscale non-rigid alignment strategy incorporating BVs and neurons.** CaliAli's multiscale registration approach ensures that both large-scale structures and finer details are accurately aligned. **a** The projections (left) utilized at each level of registration and their corresponding displacement grids (right), with magnitudes amplified 10-fold for clear visualization. **b** Results obtained utilizing a neuron-only registration strategy (left) as compared to using both BVs and neurons (right). Methods solely based on neurons can inadvertently lead to suboptimal alignment due to the juxtaposition of different yet overlapping neurons, predisposing the algorithm toward local minima. On the other hand, CaliAli starts the alignment with BVs to ensure better overlap of corresponding neuron pairs, after which neuron projections can further refine the alignment. **c** Histogram of the change in true neuron pair distances across each level of registration obtained from simulated Ca²⁺ imaging videos. At each registration level, CaliAli minimizes neuron distances to a greater

extent than using only neuron shape. **d** Example of BV and neuron projections and raw field of view (FOV) from four dentate gyrus imaging sessions acquired over 67 days. The full 7-session dataset spanning 99 days is shown in Supplementary Fig. 3a. **e** Average spatial correlation and crispness score of aligned neuron projections after registration with different algorithms from existing Ca²⁺ imaging packages. Each algorithm is listed with the projection used for inter-session motion correction. Repeated measures one-way ANOVA with Geisser-Greenhouse correction and Dunnett's multiple comparison test (each vs. CaliAli). *$p < 0.05$, **$p < 0.01$, ***$p < 0.001$. $n = 7$ sessions. Error bars = standard error of the mean (SEM). **f** Example of aligned neuron projections from different session pairs using different Ca²⁺ imaging packages: CellReg (Demons, neurons), MIN1PIPE (LogDemons, neurons), and CaImAn (patch phase correlation, filtered mean). Source data are provided as a Source Data file.

performance when BV similarity scores were above a threshold of 2.7; however, neuronal tracking performance was markedly reduced below this threshold (Fig. 3d). In a DG imaging experiment, values above this threshold were maintained for over 90 days if the focal plane was dynamically adjusted and over 40 days if the focal plane was fixed (Fig. 3a).

## Weighted groupwise alignment strategy for robust multi-session alignment

Although 90 days would surpass most ambitious tracking experiments, the declining BV similarity score over time implies that unlimited alignment may not be feasible under certain recording conditions. However, with more than two sessions, establishment of a shared

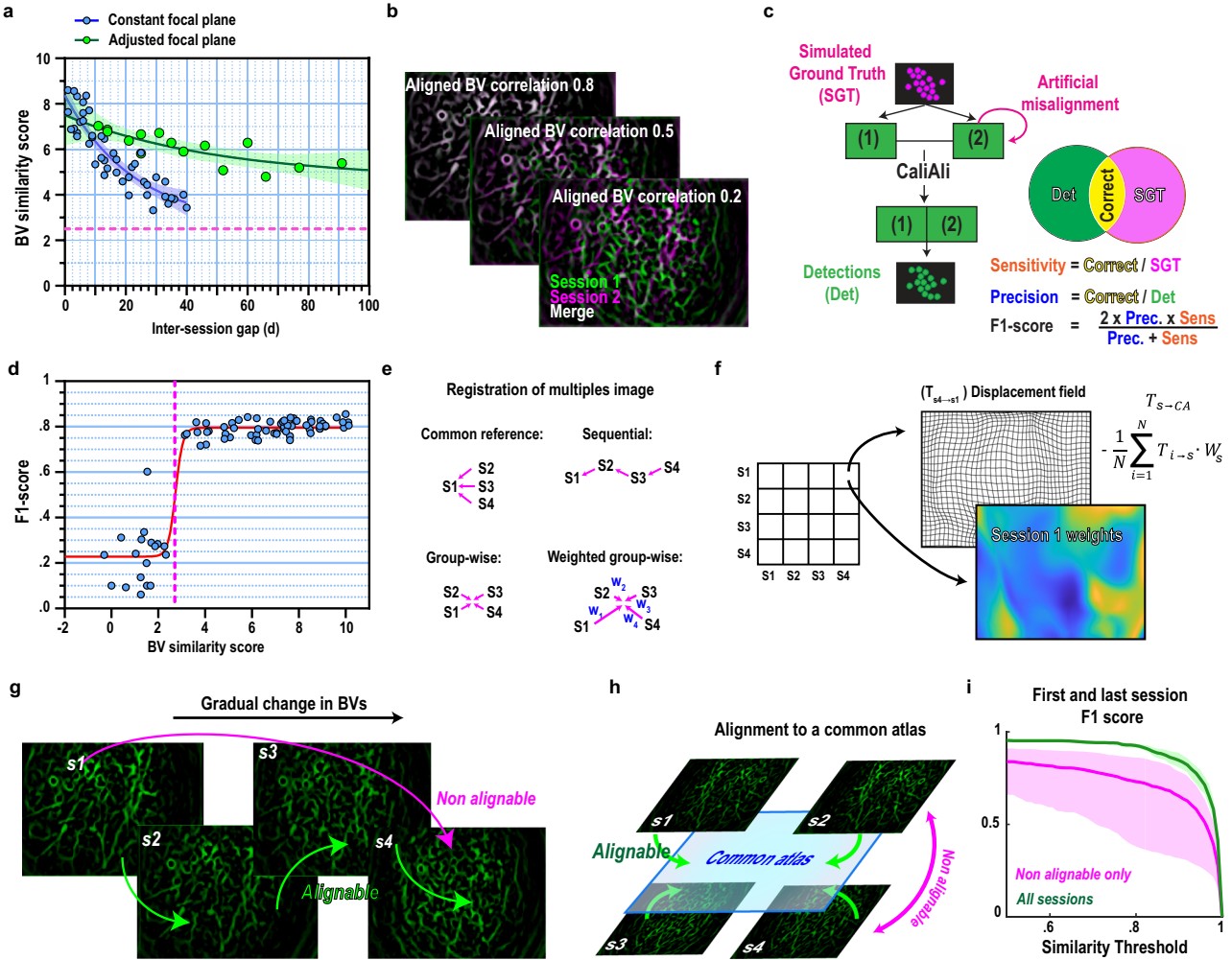

**Fig. 3 | A weighted group-wise multi-session alignment strategy ensures accurate session alignment despite significant variations in the FOV. a** BV similarity scores across different inter-session intervals (Constant focal plane, $n = 46$ session pairs; Adjusted focal plane, $n = 15$ session pairs). Fit is an exponential function. **b** BV projections bestowing different levels of spatial correlation across sessions were simulated to examine CaliAli performance. **c** Simulated $Ca^{2+}$ imaging videos (in which the identity of each neuron is known) were systematically mis-aligned. CaliAli was used to align and extract the neuronal signals from these videos. Extracted neuronal components by CaliAli were compared with the simulated ground truth (SGT) to estimate performance metrics. **d** CaliAli's performance in relation to different levels of BV stability. Red line reflects a sigmoidal fit (Hill equation), and dashed line indicates the threshold calculated as the log sigmoidal

midpoint. This threshold line is also shown in (**a**). **e** Possible strategies to register more than two sessions. **f** Implementation of a weighted group wise alignment strategy. **g** Simulation of a scenario in which BVs gradually drift in time, such as that when consecutive sessions but not distant sessions are able to be aligned. **h** A weighted group wise strategy finds a common atlas into which all sessions can be aligned, even if some session pairs are not aligned. **i** Tracking performance between the first and last session with or without intermediate sessions ($n = 8$ video simulations). In **d** traces obtained with CaliAli were considered as a true-positives if they had a temporal similarity with a SGT neuron > 0.8. In **i** all similarity thresholds are shown. In all panels data are presented as mean +/− 95% confidence interval obtained by bias-corrected and accelerated (BCa) bootstrap. Source data are provided as a Source Data file.

coordinate system could allow alignment even if certain session pairs cannot be aligned individually. Multi-session alignment is commonly addressed by employing one of two strategies: aligning each session to a common reference session or sequentially aligning each session with its predecessor (Fig. 3e, top)[2,3]. However, consistent changes in the FOV make it difficult to select a suitable reference, and sequential alignment is vulnerable to error propagation because each transformation depends on the previous one.

An alternative approach is groupwise alignment methods in which each session is aligned to an atlas that minimizes displacement across all sessions. Transformation into this atlas can be achieved by calculating the average displacement field across all session pairs (Fig. 3e, bottom left). CaliAli advances this method by calculating an atlas based on both minimal displacement and structural similarities across sessions (Fig. 3e, bottom right), as features in each session may not be equally informative for alignment. This is achieved by iteratively

weighting transformations based on the similarity of aligned session pairs (Fig. 3f). Formally, the alignment of session $s$ to the common atlas is given by:

$$T_{s \to CA} = -\frac{1}{N} \sum_{i=1}^{N} T_{i \to s} \cdot W_s \qquad (1)$$

where $T_{i \to s}$ is the displacement field that aligns session $i$ with session $s$, and $W_s$ is a weight matrix reflecting the average post-alignment similarity of session $s$ with each other session. In practice, the weight matrix is calculated by estimated local BV structural similarity[26] within a radius of 5 pixels, ensuring that for different features in the FOV, the most stable session is prioritized.

To evaluate the advantages of this approach, we simulated imaging sessions in which the BV correlation changed progressively over time, making two distant sessions impossible to align directly (Fig. 3g).

By creating an atlas allowing alignment across all sessions (Fig. 3h), CaliAli showed a notable enhancement in neuronal tracking performance, even between sessions that could not be aligned directly (Fig. 3i).

## CaliAli benefits from but does not depend on BVs for inter-session alignment

The relationship between neurons and vasculature is well-documented in studies focusing on neurovascular units[16]. However, the visibility of BVs can sometimes be compromised in certain $Ca^{2+}$ imaging setups. This reduced visibility often arises in low-contrast recording settings due to factors such as weak $Ca^{2+}$ sensors, short exposure times, or suboptimal focal planes. Therefore, CaliAli assesses the usefulness of BVs through the BV similarity score. In cases in which BVs are not prominently visible, a Hessian filter primarily detects noise, leading to a decline in BV similarity score between sessions. Should this score dip below a threshold of 2.7 (as established from Fig. 3d), CaliAli automatically switches to a neuron-only alignment mode, also issuing an alert to the user. Through simulations in which BV visibility was systematically reduced, we observed CaliAli's robust performance even when BV visibility was reduced by up to 50% (Supplementary Fig. 5). Performance started to wane slightly beyond this, as CaliAli began transitioning to a neuron-only alignment mode. This adaptive approach allows CaliAli to harness the structural details of BVs when available but not be entirely dependent on them.

## Dual-color imaging validates CaliAli's improved neuronal alignment

The primary challenge in tracking neurons with one-photon $Ca^{2+}$ imaging stems from variations in active neuronal populations across sessions. To overcome this, we applied CaliAli to data from a previous study in which neurons co-expressing GCaMP (green) and the stable nuclear marker tdTomato (red) were imaged with a dual-channel miniscope[27]. Unlike GCaMP signals that fluctuate with neuronal activity, tdTomato emits stable red fluorescence, allowing continuous visualization of neuronal somata even when cells are inactive (Fig. 4a, b). This dual-color strategy has proven to be a reliable method for tracking neurons across sessions[27]. We matched neurons identified in both channels to establish near ground truth identification across sessions (Fig. 4c). Then, we compared CaliAli's performance to other algorithms in aligning these neuronal footprints using only GCaMP signals (Fig. 4d–e). CaliAli minimized the centroid distance of neurons identified in the red channel significantly more than other approaches, demonstrating improved neuronal alignment across sessions.

## Incorporating BVs improves the detectability of dissimilar FOVs

Similar to other tracking algorithms[2], CaliAli detects when neurons in the FOV do not show resemblance across sessions, which may be useful for assessing situations of substantial FOV change. However, unlike other alignment methods, CaliAli also incorporates BVs, which may prevent the alignment of different neurons together and over-estimation of FOV stability (Supplementary Note 1; Supplementary Fig. 6).

## CaliAli detrends video sessions to facilitate processing of concatenated videos

After correcting inter-session misalignments, CaliAli concatenates video sessions into a single video sequence from which neuronal signals are extracted. However, artifacts may be introduced at the concatenation point between sessions (Supplementary Fig. 7a), which can hinder subsequent neuronal extraction. To address this, CaliAli detrends and scales the noise in each pixel's fluorescence signal prior to concatenation (Supplementary Fig. 7b, c). In addition to pixel standardization, CaliAli optionally implements background subtraction and neuronal enhancing modules proposed by Lu et al.[20] to address more complicated background fluctuations (Supplementary Fig. 7d).

## CaliAli's neuronal extraction pipeline is optimized for concatenated videos

To accurately extract denoised and demixed $Ca^{2+}$ signals from one-photon data, it is necessary to model the entire spatiotemporal structure of the data. This task cannot be achieved by simply segmenting a static image, such as a motion-corrected maximum projection or correlation image. CaliAli tackles this challenge by customizing the Constrained Nonnegative Matrix Factorization (CNMF)-E[19] pipeline to better handle concatenated videos from multiple sessions. The original CNMF-E pipeline involves two steps: initialization and matrix factorization. In the initialization step, candidate neuron locations (i.e., seed pixels) are derived from correlation and PNR images (Fig. 5a, left). Rough estimation of the spatial and temporal components of neurons is performed by analyzing the correlation of seed pixels with their neighbors. Next, the CNMF module iteratively updates spatial and temporal components to demix and denoise signals from the initialized components while also accounting for intricate background fluctuations in one-photon $Ca^{2+}$ imaging[19] (Fig. 5a, right). CaliAli introduces the following adaptations to the initialization and CNMF modules:

**CaliAli improves neuronal detectability by calculating correlation images from small batches of data.** Video concatenation can decrease the SNR in the correlation image when the activity of target neurons changes over time, leading to reduced detectability of neurons due to signal dilution. This situation was examined by simulating a concatenated video of 50,000 frames in which a specific neuron was active only in the first 20% of frames (Fig. 5b). Although this neuron was detectable when processing the initial frames, processing more frames worsened the SNR in the correlation image (Fig. 5c). To mitigate this issue, CaliAli computes a maximum projection from correlation images obtained from shorter video segments (i.e., 3,000 frames) (Fig. 5d, e), which enhances neuronal detectability and reduces computational demands by loading only a few frames at a time.

**CaliAli reduces computation times for neuronal initialization.** Initialization is a time-consuming process, exacerbated by a larger number of frames in longer concatenated sequences. The standard methodology[19] initializes each neuron sequentially based on their PNRs followed by removal of its preliminary estimation from the video, thus enabling the extraction of lower SNR neurons. To expedite initialization, CaliAli initializes distantly located neurons in parallel (Fig. 5f), as bright neuron signals should primarily affect nearby neurons. This approach reduces execution time while locally preserving the sequential high-to-low neuronal initialization.

**CaliAli minimizes computational demands without compromising extraction performance.** Memory constraints often make it impossible to process an entire dataset at once. Thus, previous approaches process data in sequential batches and propagate spatial components across batches[19] (Fig. 5g). The first batch undergoes standard initialization and matrix factorization. Subsequent batches use previously identified spatial components for matrix factorization, and new components are added from the residual, which is obtained by subtracting the neuronal signals of already identified components, leaving only undetected components (Supplementary Fig. 8a). However, the residual often includes non-somatic signals (i.e., dendrites) that could display a high spatial correlation and PNR and be incorrectly initialized as new neurons. These errors would further propagate as subsequent batches are processed (Supplementary Fig. 8b-c). Although negligible with few batches, this accumulation can significantly degrade performance with multiple batches (e.g., more than five) (Supplementary Fig. 8c). CaliAli addresses this issue by internally batch-processing

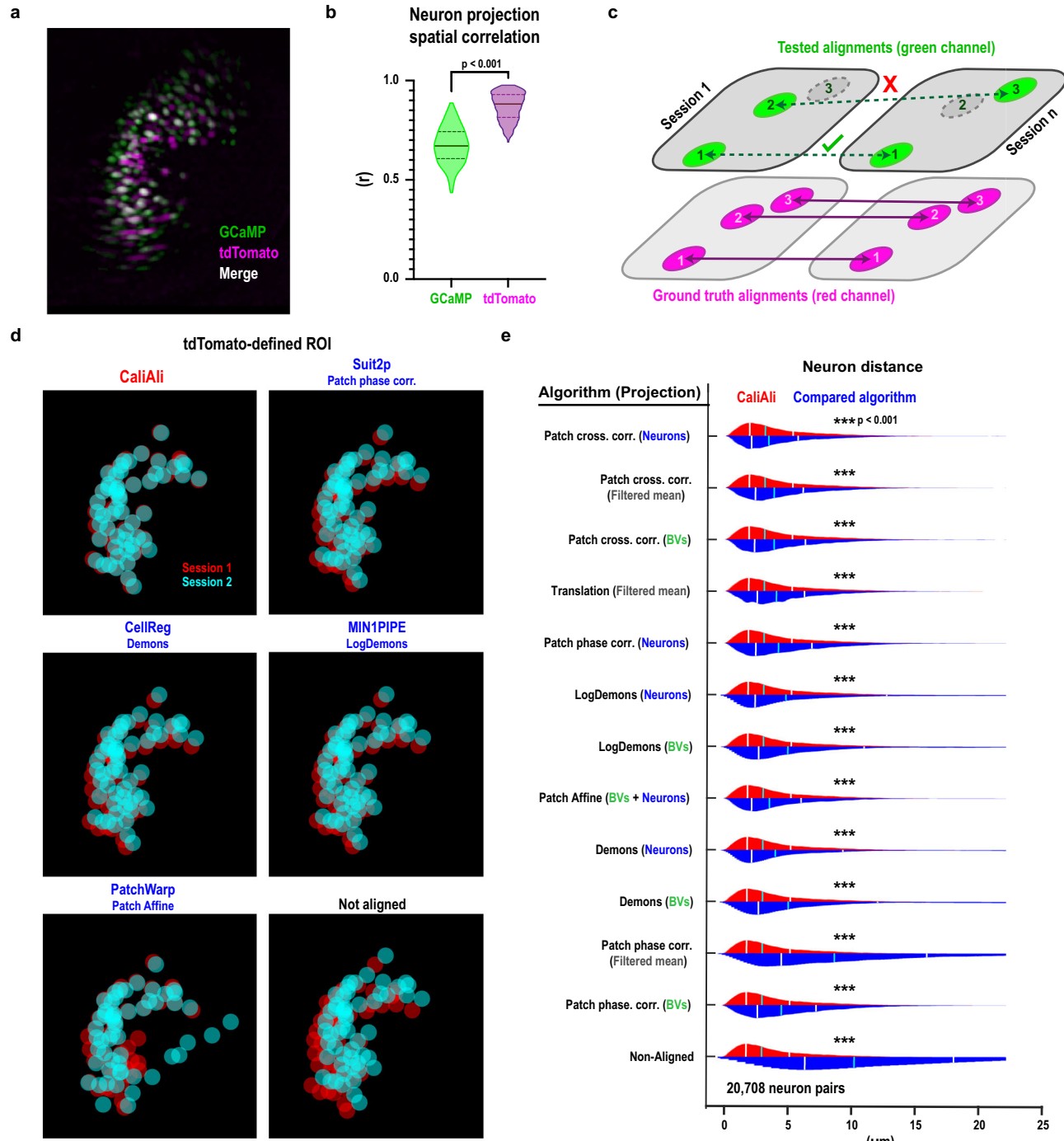

**Fig. 4 | Dual-color microendoscopy confirms CaliAli's accurate neuron registration. a** Neuron projections obtained from tdTomato- and GCaMP-corresponding channels using a dual-color microendoscope. **b** Spatial correlation of tdTomato and GCaMP projections at different time points. Data correspond to 218 session pairs obtained across eight mice within a period of 16 days. Two-sided paired t-test, t(217) = 43.46. **c** Method to evaluate alignment performance. Neuron pairs were identified by their tdTomato signal. Alignment, calculated solely from green channel data, was assessed based on the proximity of tdTomato-defined ROIs after alignment. **d** tdTomato-defined ROIs after alignment with different algorithms (name of package and default algorithm is shown). **e** Alignment performance of different algorithms. The projection used for alignment is shown in parentheses. Two-sided two-sample Kolmogorov-Smirnov test with Šidák correction for multiple comparisons. ***$p < 0.001$. Bars represent 25th, 50th, and 75th percentiles. Source data are provided as a Source Data file.

initialization and matrix factorization and producing a unified output for all batches (Fig. 5h). This is done by summarizing the components obtained in each batch; background variables are averaged (Fig. 5i, noted as W), the constant baseline image is expressed as the minimum projection across all sessions (Fig. 5i, noted as B0), spatial components are weighted and averaged based on the squared mean temporal activity of each batch (Fig. 5j), and temporal components are

concatenated (Fig. 5k). This approach prevents signal dilution by weighting the spatial component in proportion to the activity level in each session. To compare our approach with existing methods under conditions prone to high error propagation, we simulated 40 consecutive imaging sessions (5 min each). We then analyzed this concatenated data using two methods: CNMF with sequential batch processing (CNMF$_{batch}$) and standard CNMF processing of the entire

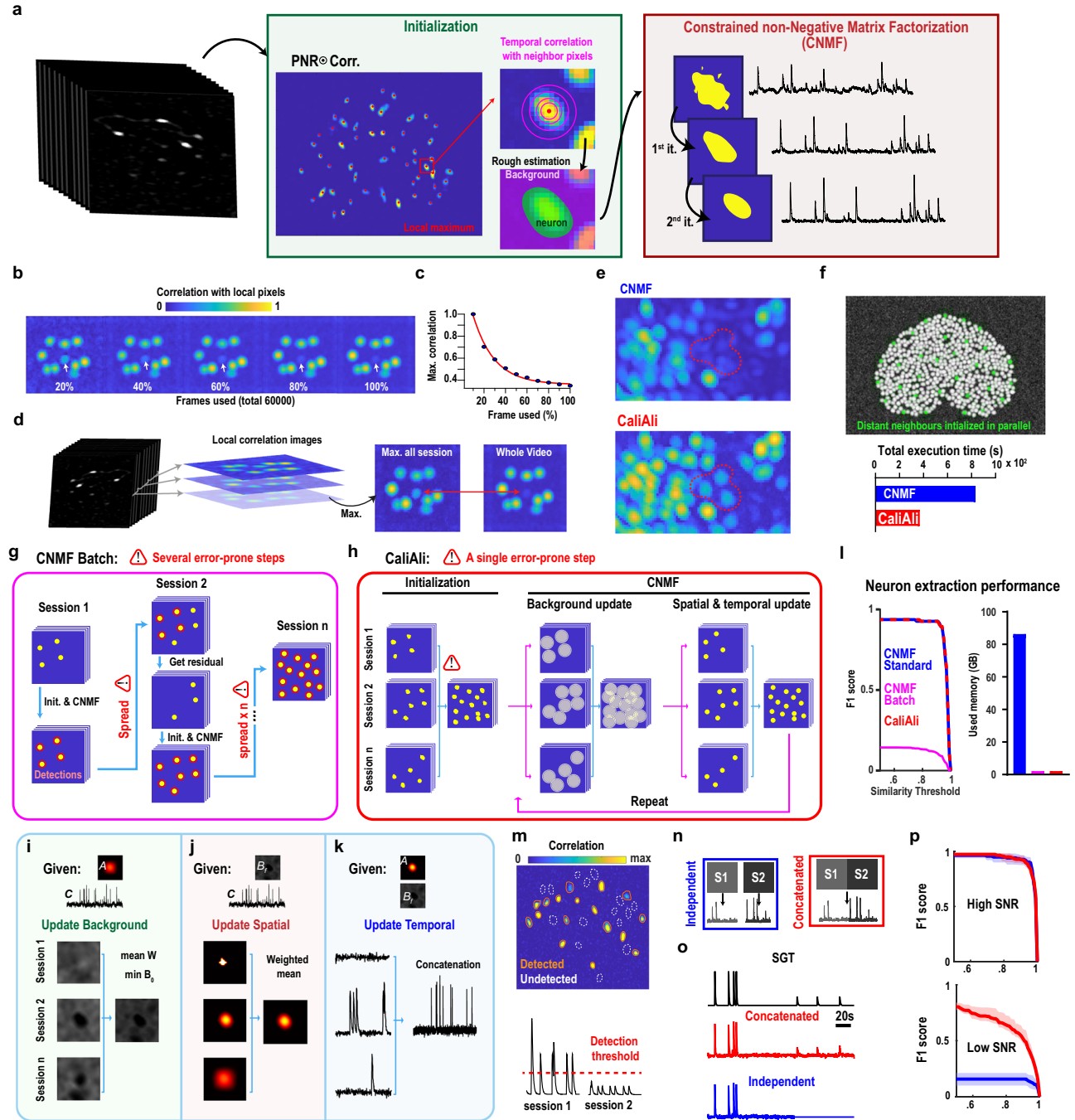

**Fig. 5 | CaliAli's extraction module enhances the detection of Ca²⁺ transients with low SNRs using a memory-efficient strategy. a** Ca²⁺ signal extraction via CNMF can be broken down into two main steps: (1) initialization (green), to estimate neuron locations; and (2) matrix factorization (red), to refine these estimates, separate overlapping components, and denoise Ca²⁺ traces. **b** Correlation image calculated from different fractions of the total video length when a neuron (white arrow) is only transiently active. **c** Maximum signal in the correlation image for the neuron shown in (**b**). **d** CaliAli computes a correlation image for each session independently and then calculates the maximum projection across all sessions. **e** Comparison of correlation images from CaliAli and standard CNMF in concatenated DG recordings. **f** CaliAli initializes distant neurons simultaneously, reducing run times compared with CNMF (12 patches processed in parallel). **g, h** Schemes depicting the batch implementations

of CNMF and CaliAli. **i–k** Panels illustrate how components are summarized across sessions. **l** Performance and memory metrics of CaliAli, CNMF, and CNMFbatch for a simulated 40-session video stack (180 × 260 pixels; 3000 frames/session). **m** Simulated scenario in which the SNR of some neurons fluctuates into undetectable levels. The upper panel shows the correlation image with dashed contours demarcating low-SNR neurons. The lower panel shows a Ca²⁺ trace with fluctuating SNR across sessions. **n** Performance was measured by comparing signal extraction from concatenated stacks vs. independent session-based extraction. **o** A comparison of traces from concatenated versus independent extractions with SGT. **p** F1 score for the extraction of high- and low-SNR traces. In all panels data are presented as mean +/− 95% confidence interval obtained by BCa bootstrap (*n* = 8 video simulations). Source data are provided as a Source Data file.

video sequence at once (CNMF_{standard}). CaliAli performed on par with CNMF_{standard} in terms of F1 score and was as efficient as CNMF_{batch} in terms of memory usage, confirming the scalability of CaliAli for large data (Fig. 5l).

## CaliAli processing of concatenated video improves detection of low-SNR transients

The main advantage of extracting neuronal signals from concatenated video sessions is that the spatial information of neurons is shared across

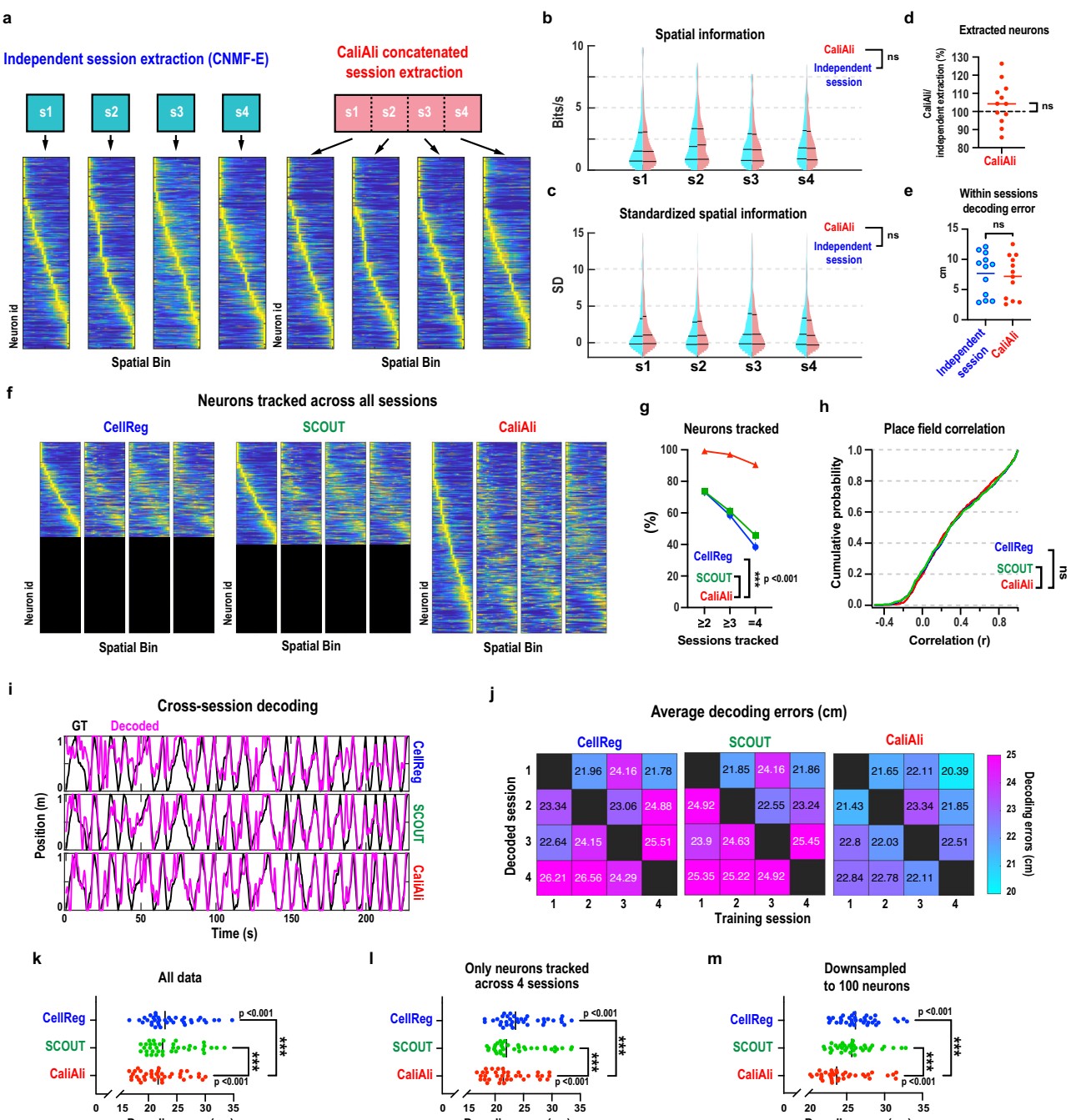

**Fig. 6 | CaliAli improves inter-session place cell stability. a** CA1 place cell activity maps from independent neuron extraction (single session) and session-aligned concatenation (CaliAli). In each session, neurons were sorted according to the location of the peak Ca²⁺ activity rate. **b** Spatial information of individual neurons obtained from a single session and CaliAli. Mixed-effects model with average neuron activity as a covariate to reduce bias from activity differences. Session and mouse identity included as random effects to account for nesting (neuron/session/ mouse). CaliAli vs. individual: F(1, 6432) = 0.109. **c** Same as b, but spatial information is expressed as SD above chance (random temporal shifts; 1000 replicates). CaliAli vs. individual F(1, 6432) = 0.218. **d** Number of neurons extracted by CaliAli vs. independent session processing. Two-sided one-sample t-test, t(11) = 1.3; n = 4 sessions/3 mice. **e** Within-session decoding of mouse position utilizing linear

support vector regression. Two-sided unpaired t-test, t(22) = 0.34; n = 4 sessions/3 mice. **f** Place cell activity maps sorted by place field locations in the first session. Only neurons with detected activity across all four sessions are shown. **g** Percentage of neurons tracked across sessions. Generalized linear mixed-effects model (binomial distribution) with session/mouse identity as random effects; F(2,13873) = 165.91. **h** Distribution of average place field correlations for each neuron pair. Two-sided two-sample Kolmogorov-Smirnov test. n = 822, 925 and 815 neurons for CellReg, SCOUT and CaliAli. **i** Across-session decoding of a mouse trajectories. **j**–**m**, Decoding root-mean-square errors across session pairs. Repeated measures one-way ANOVA (k, F(2, 70) = 18.72; l, F(2, 70) = 18.91; m, F(2, 70) = 14.51) with Holm- Šídák's correction for multiple comparisons. n = 12 session pairs/3 mice. ***p < 0.001. Vertical bars = mean. Source data are provided as a Source Data file.

sessions, allowing detection of Ca²⁺ signals that would be otherwise missed if sessions were processed independently. To examine this possibility, we simulated videos in which neurons exhibited signals that were discernible in only one of two sessions (Fig. 5m). Through its

concatenation strategy, CaliAli enhanced the extraction of low-SNR traces without impacting the extraction of high-SNR traces (Fig. 5n–p).

Furthermore, we demonstrated that CaliAli shows robust neuronal extraction performance against changes in neuron shape

(Supplementary Note 2; Supplementary Fig. 9) and error spread caused by shift in the focal plane (Supplementary Note 3; Supplementary Fig. 10) across sessions.

## CaliAli incorporates a guided post-extraction pipeline

CaliAli introduces an intuitive interface for identifying false-positive extractions (Supplementary Figs. 11a–c) by ranking components based on their spatial congruence, favoring consistent somatic shapes over varied non-somatic shapes (Supplementary Figs. 11d, e). This method is akin to semi-supervised clustering, where manually selecting a handful of non-neuronal components refines the list, ranking potential false-positives last. This system speeds up post-processing and is adaptable across various preparations without needing specific training data.

## CaliAli improves long-term neuronal trackability in silico

We next compared CaliAli's performance with that of other popular neuronal tracking algorithms (CellReg[2] and SCOUT[3]) in three scenarios common to Ca²⁺ imaging experiments: (1) low overlap of neurons with active populations remaining consistent across sessions (Supplementary Fig. 12a), (2) high overlap of neurons (Supplementary Fig. 12b), and (3) high remapping of neuronal population activity (Supplementary Fig. 12c). In simulations, we considered an amount of non-rigid misalignment equal to that seen in vivo over 4 days (Supplementary Fig. 12d). CaliAli performed better than other methods in all scenarios (Supplementary Figs. 12a–c), indicating that it improves trackability in both ideal and challenging conditions.

We also simulated a more complex scenario in which two orthogonal neuronal representations gradually change over time while preserving information content (Supplementary Fig. 12e), a common phenomenon in the hippocampus[28]. Dimensionality reduction[29] and unsupervised clustering[30] of Ca²⁺ traces obtained by CaliAli recapitulated the multi-dimensional structure of neuronal trajectories in a more precise manner than other methods (Supplementary Figs. 12f, g).

## CaliAli enhances the spatial coding accuracy of CA1 place cells

We re-analyzed hippocampal CA1 place cell imaging data from a previous study[3] to determine whether CaliAli's concatenation of videos affected neuron extraction. Our analysis revealed no significant difference in place cell coding stability between concatenated and independently analyzed sessions (Fig. 6a, b). To further address potential bias, we included average neuronal activity as a covariate or expressed information content as standard deviation above chance (Fig. 6c). This controlled for CaliAli's ability to extract low-activity neurons, but still no significant difference was observed between the two approaches. Furthermore, the number of neurons detected by CaliAli (Fig. 6d) and the accuracy of position decoding within each session were comparable whether sessions were processed independently or concatenated (Fig. 6e). However, CaliAli tracked a large proportion of neurons (Fig. 6f, g), which exhibited similar place field correlations as those tracked by other methods (Fig. 6h). Notably, CaliAli outperformed other methods when decoding animal position using training and testing data from separate sessions (Fig. 6i–m). This performance advantage persisted regardless of whether all neurons were included in the analysis (Fig. 6k), only tracked neurons were considered (Fig. 6l), or datasets were downsampled to 100 neurons to ensure equal sample sizes across methods (Fig. 6m). Taken together, these results demonstrate that CaliAli enhances the extraction of longitudinal information across multiple sessions without compromising the accuracy of within-session analysis.

## CaliAli improves short-term neuron trackability

To test neuronal trackability across short time scales, we recorded continuous Ca²⁺ activity for 1 hour, establishing this as a pseudo-ground truth (pGT) dataset. We then segmented this recording into sessions lasting 5 or 15 min each. From these segmented sessions, we extracted and tracked neuron signals across sessions using CaliAli, CellReg, or SCOUT (Supplementary Fig. 13a). We then compared the reconstructed signals from these sessions with the pGT (Supplementary Fig. 13b, c). Notably, signals obtained with CaliAli exhibited greater similarity to the pGT dataset compared with other methods, irrespective of the duration of imaging sessions (Supplementary Figs. 13d, e).

## CaliAli improves long-term neuron trackability as determined by optogenetic tagging of neurons

To establish a ground truth for evaluating neuron tracking, we employed opto-tagging to identify and follow specific neurons across imaging sessions. Specifically, we transduced the Ca²⁺ sensor GCaMP6s and stimulatory opsin ChrimsonR into DG neurons (Fig. 7a), resulting in a subset of GCaMP6s-expressing neurons activated by ChrimsonR upon orange light stimulation (Fig. 7b). If neuronal tracking is accurate, neurons responding to light stimulation in one session would be expected to respond again in a subsequent session (Fig. 7c, opto-consistent neurons). To evaluate this, we imaged opto-evoked responses across two sessions separated by 30 min. CaliAli showed a higher proportion of opto-consistent neurons among the responsive population compared with other methods (Fig. 7d). Introducing incremental misalignments to the videos gradually decreased opto-consistency, confirming that the observed opto-consistency was due to accurate neuron alignment and not chance (Fig. 7e). Note that to ensure accurate extraction of somatic signals during simultaneous imaging and optogenetic manipulation, sparse expression of Chrimson and GCaMP was employed. We validated the somatic origin of signals extracted by CaliAli using two independent approaches: comparing the shapes of signals to those obtained with CNMF-E from independently processed sessions and utilizing a machine learning model specifically designed to unbiasedly quantify neuropil signals and differentiate them from somatic signals (Supplementary Note 4; Supplementary Fig. 14).

We then extended our analysis to a longer timeframe of 4 weeks. We found that a minority of extracted components responded to light stimulation (32.4% ± 3.4), indicating that achieving high opto-consistency through inadvertent alignment of different neurons is unlikely. Consistent with observations from shorter inter-session intervals, CaliAli consistently exhibited higher opto-consistency than other methods (Fig. 7f–h). Notably, CellReg's low performance was likely due to the sparse GCaMP expression required for these experiments, which challenges registration algorithms relying solely on neuronal shapes for motion correction (Supplementary Fig. 3d). SCOUT may be less affected by this limitation, as it incorporates temporal metrics for neuron tracking.

## CaliAli-tracked neurons exhibit more stable population activity for up to 99 days

We next utilized CaliAli to track DG neurons for 99 days. To evaluate the long-term stability of DG neurons across sessions, we calculated population vector distance (PVD) across sessions (Fig. 8a). Population vectors were more similar across sessions for data processed with CaliAli than with other methods (Fig. 8b, c), consistent with observations that DG neurons produce stable neuronal representations over time[31]. For PVD calculation, neurons not detected in all sessions were excluded to ensure equal dimensions across samples. Notably, assigning zero activity to these undetected neurons further amplified the performance gap between CaliAli and other methods (Supplementary Figs. 15a, b).

## CaliAli captures gradual and continuous drift in DG neuronal representations

Neuronal representations are known to gradually drift over time[32]. To investigate the drifting of DG population activity, we plotted PVD

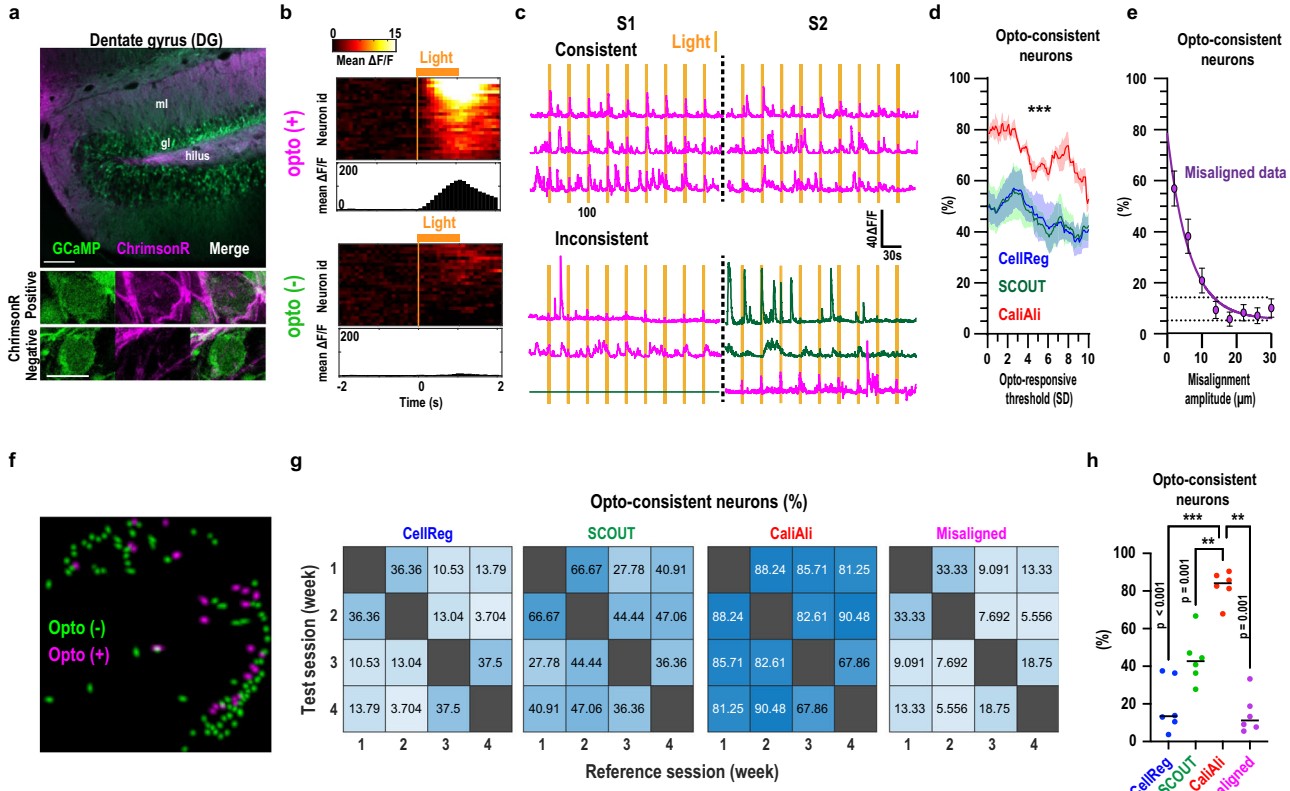

**Fig. 7 | Neurons tracked using CaliAli show more consistent optogenetic responses than neurons tracked by other methods. a** Histology of gCaMP and ChrimsonR. Scale bars = 100 μm and 5 μm. ml = molecular layer; gl = granule cell layer. ChrimsonR expression is only evident at higher magnifications (corroborated in at least 3 independent experiments). **b** Heatmap and average peristimulus time histograms of light-responsive (opto(+)) and non-responsive (opto(-)) neurons. Neurons showing light-evoked responses 3 standard deviations (SD) above baseline were considered opto(+). **c** Representative traces for opto-consistent (i.e., responded to light stimulation in both sessions, S1 and S2) and -inconsistent neurons (i.e., responded in only one session). Magenta = opto(+), green = opto(-). **d** Percentage of opto-consistent neurons for different opto(+) thresholds (for i, the threshold is 3 SD). Generalized linear mixed-effects model (binomial distribution) with a random effect accounting for variation in CNMF initialization parameters;

F(2, 71,859) = 136.24. Shaded areas represent variations due to parameter selection, showing 95% confidence intervals (BCa bootstrap, 1000 replicates) across 8 different CNMF initialization parameters applied to data from one mouse.
**e** Percentage of opto-consistent neurons remaining after increasing misalignment was applied to CaliAli-aligned data. The solid purple line is an exponential fit to the data. Dashed lines indicate chance level performance. Error bars = SEM. **f** Footprint projection of opto(+) and opto(-) neurons. **g** Percentage of opto-consistent neurons across session pairs obtained by different tracking methods. For the misaligned group, Ca²⁺ signals were extracted after introducing a 15μm misalignment. **h** Percentage of opto-consistent neurons across all session pairs. Repeated measures one-way ANOVA with Dunnett's multiple comparisons, F(1.4, 7.0) = 16.5, n = 6 session pairs. ***p < 0.001, **p < 0.01. Horizontal bars = mean. Source data are provided as a Source Data file.

against the interval between sessions (Supplementary Fig. 15c). Population vectors tracked by CaliAli exhibited linear drift over time, contrasting with the exponential drift observed with CellReg, SCOUT, or non-aligned data.

To explore the nature of these temporal dynamics, we plotted PVD from non-aligned data against the degree of inter-session misalignment (Supplementary Fig. 15d). Remarkably, we found an exponential rather than linear relationship between PVD and misalignment amplitude, suggesting that the linear drift of PVD detected by CaliAli cannot be explained by gradual misalignment across sessions. This also shows that misalignment amplitudes decrease in proportion to PVD, reinforcing that PVD reflects alignment performance. Overall, these results suggest that CaliAli uniquely captures the gradual and continuous drift in DG neuronal representations, a phenomenon not discerned by other methods.

## Discussion

In this study, we introduce CaliAli, a comprehensive suite for longitudinal neuronal signal extraction in multi-session one-photon Ca²⁺ imaging experiments. CaliAli possesses unique properties that distinguish it from other tracking strategies (Supplementary Table 1). CaliAli facilitates the long-term tracking of neurons owing to two main

contributions: (1) a multiscale approach integrating neurons and BVs to correct inter-session alignment and (2) a Ca²⁺ extraction pipeline customized for processing multi-session concatenated videos. These functions are integrated into a comprehensive computational suite, enabling extraction of long-term tracked neuronal signals from raw multi-session imaging data.

CaliAli's multiscale inter-session alignment uniquely addresses the complexity of FOV misalignment observed across large temporal scales, distinguishing it from earlier implementations that are primarily adapted from motion correction algorithms designed for within-session alignment. CaliAli corrects inter-session misalignment even when active neuron populations change across sessions. Thus, CaliAli is ideal for studying brain processes such as representational drift[5,12,32] or activity remapping[7,8]. Indeed, our simulations demonstrate that CaliAli enables more accurate clustering of drifting activity than other methods. Furthermore, CaliAli uniquely captures the gradual and continuous drift of DG population vectors over a 99-day period.

CaliAli possesses a computationally efficient neuronal extraction pipeline designed for processing concatenated multi-session data to enhance the detection of low-SNR Ca²⁺ transients. This improved detectability is crucial for deciphering brain dynamics reliant on sparse coding, through which neurons with low or negligible responses

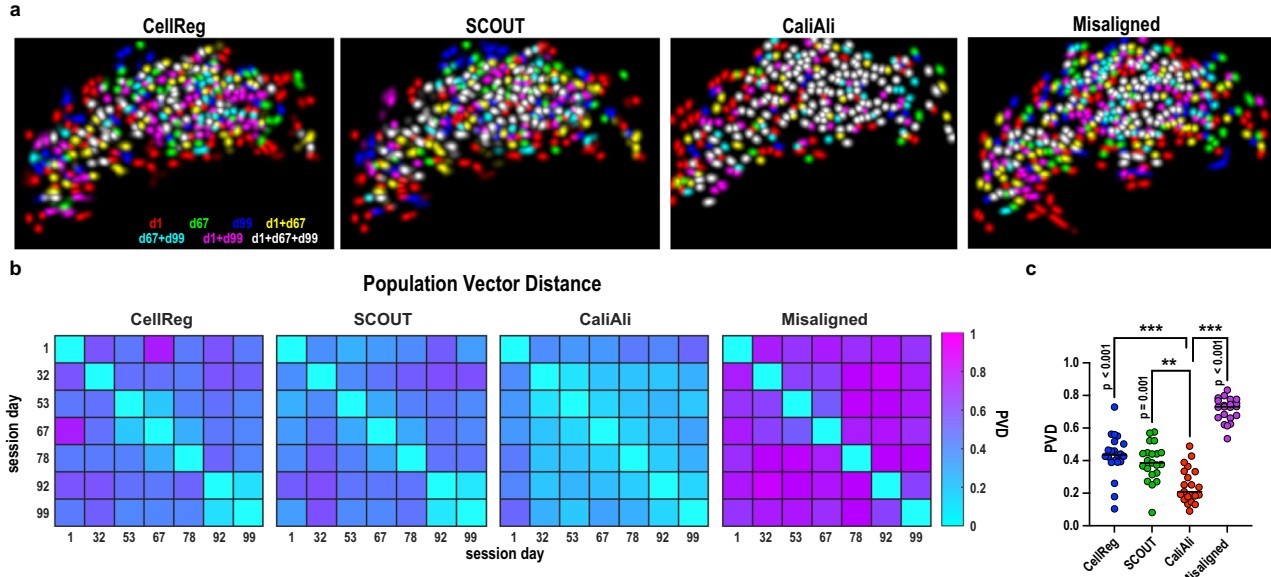

**Fig. 8 | Neurons tracked using CaliAli show higher intra-session population vector stability than neurons tracked by other methods. a** Overlay of the spatial components of detected neurons on days 1, 67, and 99 for different tracking methods. **b** Population vector distance (PVD) of the average activities of neurons were calculated across different sessions. A higher PVD reflects greater variation in average neuronal activity across sessions. **c** PVD across all session pairs. Repeated measures one-way ANOVA with Dunnett's multiple comparisons, $F(2.4, 48.3) = 106.6$, $n = 21$ session pairs. \*\*\*$p < 0.001$, \*\*$p < 0.01$. Horizontal bars = mean. Source data are provided as a Source Data file.

contribute to the encoding of information[33]. Indeed, in some brain regions, information coding is inversely related to neuron activity rate[34]. Therefore, CaliAli is an optimal tool for studying this sparse activity in multi-session experiments.

CaliAli also guarantees a consistent number of neurons tracked across sessions, which is crucial for applying dimensionality reduction and ensemble detection algorithms in multi-session data (e.g., PCA[35], tensor component analysis[36], functional clustering[37], non-negative matrix factorization[9]). Utilizing place cell data, we found that CaliAli increased the number of neurons that could be tracked across four sessions. These additionally tracked neurons provided equal or higher spatial information than neurons detected by other approaches, showed by the improved decoding accuracy of mouse position. Increasing the number of trackable neurons is essential for studying long-term brain dynamics, as the collective activity of large neuronal populations often exhibits complex multidimensional patterns that cannot be predicted from the activity of individual neurons[38].

Although offering significant benefits to long-term trackability, CaliAli has some trade-offs in terms of portability compared with tools like CellReg. This is because each module within CaliAli (e.g., pre-processing, motion correction) requires optimization specifically for concatenated video processing. Consequently, seamlessly integrating CaliAli modules with other implementations necessitates minor adjustments to ensure compatibility. This also implies that utilizing CaliAli with data from past studies requires reprocessing data directly from raw videos. Also, the better tracking performance achieved by CaliAli may be predominantly evidenced in challenging neuronal tracking conditions (i.e., high neuronal overlap, remapping of active neurons across sessions, complex non-rigid inter-session misalignment) that may particularly occur in long-term tracking experiments. Indeed, when assessing trackability through optogenetic-tagging, CaliAli demonstrated tracking enhancements over the course of several weeks.

In summary, CaliAli is a powerful tool that excels in diverse neuronal tracking scenarios. It identifies a consistent number of tracked neurons and improves the detectability of smaller $Ca^{2+}$ transients,

making it suitable for studying brain dynamics over long periods of time, including representational drift, remote memory processing, neuron development, and behavior measured across multiple sessions.

## Methods
### Animals
All animal experiments were approved by the University of Tsukuba Institutional Animal Care and Use Committee. Mice were kept in a home cage (260 × 232 × 210 mm) in an insulated chamber maintained at an ambient temperature of 23.5 ± 2.0 °C with a 12-h light/dark cycle and ad libitum access to food and water according to institutional guidelines. The details of mouse lines used through this study are provided in Supplementary Table 2. No specific allocation based on sex was performed, as the focus of this study is on algorithm development rather than biological variability.

### Virus and injections
Adeno-associated viruses were prepared as previously described[39]. Mice at 6–28 weeks of age were anesthetized using isoflurane and secured in a stereotaxic apparatus (Stoelting, USA). The injection was performed using a Picospritzer III air pressure system (S48 Stumilator, Grass Technologies, USA) connected to a glass pipette injector. The injection process lasted 10 min, after which the injector needle remained in position for 5 min before being gently removed. Following injection, mice were given a minimum recovery period of 1 week before lens implantation. Virus and stereotaxic coordinates used are detailed in Supplementary Table 2.

### Lens implantation
A microendoscope lens (1-mm diameter, 4-mm length, Inscopix, USA) was used for imaging experiments. For DG recording, the lens was placed at AP -2.0 mm, ML + 1.25 mm relative to bregma and 1.53 mm below the dura. For CA1 recording, the lens was angled at 15°, directed medially, and positioned to contact the corpus callosum above the injection site, 1.14 mm below the dura. At least 1 week after lens implantation, a baseplate was attached above the implanted lens. After

baseplate surgery, mice were habituated to a dummy microendoscope for 1-2 weeks before recording.

## Preparation of tissue sections

After imaging, mice were perfused transcardially with phosphate-buffered saline (PBS; 0.1 M) and 4% paraformaldehyde (PFA). Brains were removed, fixed overnight in PFA, and transferred to PBS. Coronal sections (30 μm) were cut using a vibratome (VT1200S, Leica). Sections were mounted on slides with mounting medium containing DAPI. Images of GCaMP6s- and ChrimsonR-Tdtomato-expressing neurons were obtained using a Zeiss Axio Observer Z1 microscope and Leica TCS SP8+ confocal microscope (Leica, Germany).

## Ca$^{2+}$ imaging and optogenetic manipulation

A miniaturized microscope with flexible light source input (T-scope V4, Physiotech Inc, Japan)[40] was utilized for neuronal imaging and manipulation. For imaging without optogenetic manipulation, we used a blue laser (473 nm, Shanghai Laser & Optics Century Co., Ltd., China) delivering 0.05-0.6 mW at the bottom of the T-scope. For the opto-tagging experiment, we used a custom-made blue laser (445 nm) or commercially available blue laser (473 nm, Shanghai Laser & Optics Century Co., Ltd., China) delivering 0.1-1.1 mW and an orange laser (589 nm, Shanghai Laser & Optics Century Co., Ltd.) delivering 0.5 mW at the bottom of the T-scope. Stimulation was delivered through a custom-made laser combiner and an optic patch cable (Thorlabs, Japan). For all experiments, images were acquired at 10 frames/s. Laser intensity, gain, and exposure settings were customized for each mouse while monitoring the fluorescence intensity histogram to ensure that the highest possible dynamic range was achieved without signal saturation. For the opto-tagging experiment, the blue laser was used to stimulate GCaMP, and the orange laser was used to stimulate ChrimsonR (1-s, 10-Hz, 50% duty cycle).

## Ca$^{2+}$ imaging experimental details

**Obtaining realistic neuronal parameters from DG imaging data and estimation of non-rigid misalignment amplitudes.** DG Ca$^{2+}$ imaging recordings were conducted with eight mice in two separate 5-min sessions with a 4-day interval between sessions. During each session, mice were placed in a chamber with white plastic walls and a stainless-steel grid measuring 310 mm in width, 250 mm in depth, and 280 mm in height. Mice were allowed to freely explore the chamber during each session. Parameters obtained from recorded neurons were used to create video simulations and were also used to estimate the inter-session misalignments shown in Supplementary Fig. 1a-c.

To estimate non-rigid misalignments over long temporal scales (Supplementary Fig. 1d) DG and CA1 Ca$^{2+}$ imaging recordings were conducted for up to 99 days in the home cage. Session duration was 5-10 min long.

**Opto-tagging experiments.** Opto-tagging experiments (Figs. 7; Supplementary Fig. 14) were carried out in the home cage. Each session consisted of a 5-min baseline recording followed by 5 min of opto-stimulation. Opto-stimulation was conducted over 10 trials, with each trial lasting 30 s. During each trial, light stimulation was provided for 1 s, beginning after 13 s. The data depicted in Fig. 7a-3 correspond to two recording sessions with a 30-min interval between sessions and in Fig. 7f–h correspond to four sessions with 7-day intervals between sessions.

**Long-term tracking of DG data.** Data for Figs. 8 and Supplementary Fig. 15 were collected from DG recordings during free exploration of the home cage, with each session lasting 10 min. Recordings were conducted on days 1, 32, 53, 67, 78, 92, and 99.

**Performance in the presence of z-axis displacement.** Data corresponding to Supplementary Figs. 6 and 10 were collected from DG recordings during free exploration of the home cage. Recordings were conducted on three different days. Each day, three consecutive 5-min sessions were recorded. Sessions 1 and 3 were carried out using the same focal plane, whereas session 2 was obtained by shifting the focal plane by 6 μm (day 1), 23 μm (day 2), or 62 μm (day 3). On each day, the two focal planes were chosen carefully to ensure that a similar number of active neuron populations could be imaged. Data associated with Supplementary Fig. 6f were obtained from the Inscopix database.

**Change in spatial components across sessions.** The spatial components used to simulate changes in neuron shapes across sessions, as depicted in Supplementary Fig. 9, were retrieved from source data shared in a previous study[5].

**Hippocampus CA1 place cell data.** The data on place cells depicted in Fig. 6 originated from source data made publicly available in a prior study[3]. Briefly, these data capture the Ca$^{2+}$ activity of CA1 neurons expressing GCaMP6f in 3 mice that underwent training on a 1-m-long linear track. The dataset encompasses four trials conducted over 4 days, with one trial per day. During each trial, mice underwent testing for 30 laps, typically completing the task within 10-15 min.

**Simultaneous dual-color calcium imaging.** Data was obtained in a previous study[27]. Briefly, these data capture the Ca$^{2+}$ activity of CA1 neurons co-expressing GCaMP6f and tdTomato in 8 mice that underwent training on a 1-m-long linear track. The dataset encompasses five to nine imaging sessions, each 15 minutes in duration, conducted over a 16-day period.

## In vivo Ca$^{2+}$ imaging data processing

Raw Ca$^{2+}$ imaging videos underwent two-fold spatial downsampling. Motion artifacts were corrected using BVs and the log-demons image registration algorithm (note that this module differs from the one used for inter-session alignment). Ca$^{2+}$ traces were extracted by CNMF-E with the modifications described in the main text. The spatial filtering (gSig) was set to 2.5. For simulations, the minimum PNR (min_pnr) and minimum pixel local correlation (min_corr) were set to 2.5 and 0.15, respectively.

In actual Ca$^{2+}$ experiments, we manually defined min_pnr and min_corr by carefully monitoring the correlation and PNR image of each recording using the preprocessing interface included with CaliAli. We also utilized a mask function to manually select the regions in the FOV where neurons were present. These functionalities are detailed in the tutorial text included with CaliAli.

In the opto-tagging experiment shown in Fig. 7d, min_pnr was set to 5 and min_corr varied from 0.4 to 0.8 in increments of 0.05. This was performed to assess the performance of the CaliAli algorithm under different scenarios, ranging from a scenario where most neurons were extracted (albeit with some false-positives) to a scenario where false-positives were minimized at the cost of potentially missing some neurons.

For concatenated data (CaliAli) and independently processed sessions (CellReg/SCOUT), false-positive extractions were manually discarded using the post-processing interface provided by CaliAli (Supplementary Fig. 11, detailed in the tutorial file included with CaliAli code), except for simulated data (where false-positive extractions were virtually null) and the data related to Fig. 7d (a range of initialization parameters were used to asses different levels of false-positives and false-negatives).

## Video simulations

Spatial components were simulated by randomly sampling from 1,137 DG neurons in eight GCaMP6f-expressing mice. Spatial components

were positioned randomly in the FOV, constrained by minimum distances to neighboring cells (low overlap: 26 μm, medium overlap: 21 μm, high overlap: 8 μm). Temporal components were simulated considering rising times produced by a Bernoulli process and subsequently convolved with a temporal kernel $g(t) = \exp(-t/\tau_d) - \exp(-t/\tau_r)$. Ca²⁺ rates and kinetics for each neuron were sampled from a lognormal distribution with parameters obtained from mouse DG recordings: transient probability μ = -4.9, σ = 2.25; $\tau_r^{-1}$ μ = 2.08, σ = 0.29; $\tau_d^{-1}$, μ = 0.55, σ = 0.44. Note that the mean transient rate in our simulation was marginally higher than that in the empirical data, as neurons were required to exhibit a minimum of one Ca²⁺ transient per session. A constant PNR of 2 was used in all simulations. Local background fluctuations were modeled using a 2D Gaussian-filtered version of the spatial components (σ = 20), with weakly correlated noise generated by applying a 2D Gaussian filter with σ = 0.5 on white noise. Inter-session misalignment was emulated using gradients of a random 2D Gaussian (σ = 60). In the experiment comparing CaliAli with other tracking algorithms (Fig. 5), we scaled the misalignment fields to achieve a non-rigid misalignment amplitude of ~8 μm. This degree of misalignment is consistent with that observed in the DG over a period of 4 days (Supplementary Fig. 12d). In experiments in which we compared the alignment performance of different projections (Fig. 2), we simulated a non-rigid misalignment amplitude of ~12.5 μm, reflecting the degree of misalignment seen in DG and CA1 data over large temporal scales (Supplementary Fig. 1d).

Remapping was simulated by rendering a subset of neurons inactive in certain sessions. A variable SNR was simulated in a similar manner as in remapping, but the amplitude of Ca²⁺ transients was reduced by 80% instead of inactivating neurons. The parameters employed in each simulation are found in the source data accompanying the main text.

**Incorporation of realistic BV structures.** We utilized frames obtained from DG recordings as a static baseline. To incorporate modest variation in the static baseline, we utilized frames obtained 4 days apart, which were manually aligned and used as a static baseline for each session. For simulations utilizing more than two sessions, additional baseline frames were created by linear interpolation. To simulate larger variations in BV (Fig. 3b), we distorted a reference baseline image, denoted as $FOV_A$, using a weighted combination of $FOV_A$ and another baseline image, $FOV_B$, obtained from a different mouse. This distortion was achieved through the equation $FOV_A \bullet w + FOV_B \bullet (1 - w)$, where w is a weight parameter $\in [0, 1]$ controlling the extent of the distortion.

**Simulated FOV lacking BVs.** We generated a smooth background intensity distribution from baseline images by computing the outer product of the mean intensity values across each row and column, resulting in a matrix capturing low-frequency intensity variations (Supplementary Fig. 5). We then applied a 2D median filter with a 25 × 25 pixel window to this matrix, further smoothing the estimated background intensity. Additionally, we added a slight random Gaussian noise to the image (STD = 0.1) to help obliterate any elongated structures that could be captured by Hessian filters. The final output was an image that closely resembled the original in terms of intensity distribution but was stripped of BVs and other intricate details. This BV-free image was incrementally merged with the original image to simulate varying degrees of BV erosion utilizing the same approach used to combine FOVs.

**Neuron tracking parameters**
We utilized default parameters for SCOUT and CellReg. In all cases, FOVs were aligned with a non-rigid approach, as all simulations involved non-rigid deformations. In some cases, the footprint

registration algorithm used by SCOUT had poorer performance than CellReg, mainly when neuronal densities were low and the active neuronal population remapped. To ensure that any differences in performance were not due to incorrect footprint alignment, we used the same alignment module as used with CellReg for SCOUT. Through these modifications, we were able to maintain or improve the performance of SCOUT.

**Registration algorithms implementation**
For translation, patch cross-correlation, patch phase-correlation we utilized the NoRMCorre[17] package (https://github.com/flatironinstitute/NoRMCorre) incorporated in CNMF-E (https://github.com/zhoupc/CNMF_E) utilizing the standard implementation for one-photon Ca²⁺ image. For affine patch registration we utilized PatchWarp[25] (https://github.com/ryhattori/PatchWarp). While originally designed for two-photon imaging, we modified PatchWarp to utilize BV and neuron projections instead of the mean image, making it suitable for one-photon data. For demons registration we utilized the imregdemons MATLAB function with an accumulated field smoothing value of two, as implemented in the CellReg package[2]. For Log-demons registration we utilized Herve Lombaert MATLAB package (https://www.mathworks.com/matlabcentral/fileexchange/39194-diffeomorphic-log-demons-image-registration) with parameters set as in the MIN1PIPE package[20]

**Data analysis**
Statistical analysis was performed in MATLAB 2022a-2023a (MathWorks, Maryland, USA), Graphpad Prism (GraphPad, California, USA), and Igor Pro 9.04 (WaveMetrix, Oregon, USA). Type I error was set to α = 0.05.

**Matching of extracted component with simulated ground truth (sGT) data.** Let $E$ denote the set of extracted components and $S$ denote the set of sGT neurons. For each extracted component $e_i$ and sGT neuron $s_j$, we computed spatial similarity $Sim_{spatial}(e_i, s_j)$ using cosine similarity. Correspondingly, temporal similarity $Sim_{temporal}(e_i, s_j)$ was calculated for each component pair using the denoised Ca²⁺ data. These similarities were organized into matrices. The spatiotemporal similarity matrix $M$ was then obtained by element-wise multiplication: $M = Sim_{spatial} \odot Sim_{temporal}$. To find the optimal assignment of extracted components to SGT neurons, we solved the linear assignment problem by permuting the rows and columns of $M$ to maximize the sum of similarities between matched components:

$$argmax_\sigma \sum_{i=1}^{n} M_{i,\sigma(i)} \qquad (2)$$

where $\sigma$ denotes a permutation and $n$ is the number of extracted components. Finally, true positives (TPs) were evaluated by considering components with a temporal similarity exceeding a threshold $\tau : TP = \left\{ (e_i, s_j) | Sim_{temporal}(e_i, s_j) > \tau \right\}$. Note that unmatched components were considered to have a similarity of zero. This approach ensured that each extracted component was matched with no more than one sGT component.

**BV similarity score.** Given two projections of BV, CaliAli computes a BV similarity score by first calculating the correlation between observed BV images $C_{obs}$. Next, it generates a set of correlation values from the same BV images after introducing random non-rigid misalignment to one of the images (100 surrogates). These misalignments are simulated by utilizing gradients of a random 2D Gaussian distribution with a standard deviation of 15. The maximum misalignments are scaled to have an amplitude of 6. From this randomly misaligned data, a collection of correlation values $C_{random}$ is obtained.

Subsequently, the BV similarity score is computed using the formula:

$$BV_{score} = \frac{C_{obs} - mean(C_{random})}{std(C_{random})} \qquad (3)$$

Therefore, the BV similarity score can be interpreted as the number of standard deviations by which the actual correlation of aligned BVs differs from a situation of random alignment.

**Defining ground truth neuron coordinates in tdTomato signals imaged by dual-color imaging.** Neuronal structures in the average red-channel frame were enhanced by anisotropic diffusion and morphological opening as described by Lu et al. (2017)[20]. The resulting projection was then maximum-filtered with 2.5-pixel radius kernel. Neuron locations were identified within this filtered image as pixels that were both local maxima and exceeded a heuristically defined intensity threshold of 0.01. This process was repeated for each session, yielding a set of putative neuron locations for each session's mean projection image. Next, to match extracted neuron coordinates across sessions, we first aligned the maximum-filtered red channel projections of each session using a combination of translation and demon registration. The alignment transformations derived for each session were then applied to the corresponding neuron coordinates extracted from that session's mean projection. To establish reliable neuron matches across sessions, we generated empirical CDFs for "different-neuron" (within-session nearest-neighbor distances) and "same-neuron" (distances between putatively matched neurons across sessions using the Hungarian algorithm) distance models in the red channel. Matches with a likelihood ratio (same-neuron likelihood divided by different-neuron likelihood) greater than 1 were accepted, generating a reference set of matched neuron pairs for each session pair in the red channel, serving as ground truth for subsequent neuron tracking.

**Dimensionality reduction and unsupervised clustering of drifting neuronal activity.** The population activity shown in Supplementary Figs. 12e–g were subjected to dimensionality reduction and clustering using the UMAP algorithm[29]. Data were binned into 10-second intervals. We employed a cosine distance metric with a minimum distance threshold of 0.1 and considered the 10 nearest neighbors. These parameter values were estimated heuristically by selecting those values that produced optimal results in ground truth data. Unsupervised clustering was performed on the reduced data using the k-means algorithm (k = 2) and squared Euclidean distance metric.

**Spatial information content.** Spatial information content was computed using the same procedures described in a previous study[3]. In brief, the central region of the 1-m-long linear track (the middle 80%) was divided into 20 bins. Spatial information was calculated using the formula $\sum p_i \lambda_i \log_2(\lambda_i)$, where $p_i$ denotes the probability of the mouse being in each bin and $\lambda_i$ denotes the ratio of the probability of firing while in the bin to the mean probability of firing.

**Space decoding with place cell data.** Mouse position was decoded using Support Vector Regression (SVR) with a linear kernel on spike data from raw Ca[2+] signals smoothed with a Gaussian kernel (standard deviation = 40). Optimal SVR parameters (box constraint = 0.74, epsilon = 0.074) were determined via cross-validation. Within-session performance was evaluated using 5-fold cross-validation averages. Across-session performance was evaluated by using one session as the training dataset and other sessions as testing datasets.

**Population vector analysis.** A population vector $v$ is defined as $v = [v_1, v_2, ..., v_N]$, where $N$ is the total number of neurons and $v_i$ is the average activity of the i-th neuron. The activity of a neuron is calculated from the predicted spike sequence S denoised from the raw Ca[2+] traces.

For analysis of population vectors related to the long-term tracking of DG neurons (Fig. 8), we computed population vectors for the first and second halves of each session. We then calculated population vector distance (PVD) by computing the difference between the mean correlation distances for intra-session pairs and inter-session pairs. This approach to calculating PVD provides a measure of the discrepancy in neuronal activity patterns within and across sessions.

For the analysis of population vectors related to imaging in different focal planes (Supplementary Fig. 6), PVD was calculated as the difference between the correlation distance of session pairs in different focal planes and the correlation distance of session pairs in the same focal plane.

**Sorting of components in term of spatial congruence.** To sort components, CaliAli first calculates the cosine distance matrix between all pairs of spatial components $X = \{x_1, x_2, ..., x_n\}$. Each spatial component $x_i$ is represented as a vector. Then, the cosine distance between two spatial components $x_i$ and $x_j$ is computed by:

$$Cosine\ distance\ (x_i, x_j) = 1 - \frac{x_i \cdot x_j}{||x_i|| \, ||x_j||} \qquad (4)$$

This operation is performed for all pairs of spatial components, resulting in a cosine distance matrix $D_c$. Next, multidimensional scaling (MDS) is employed to represent these pairwise distances in a low-dimensional space Y. MDS minimizes the following strain criterion:

$$Strain(X) = \sum_{i \neq j} (d_{ij} - \hat{d}_{ij})^2 \qquad (5)$$

Here, $d_{ij}$ represents the original pairwise distance from $X$ and $\hat{d}_{ij}$ represents the corresponding distance in the low-dimensional space Y.

Finally, for each component $y_i$ corresponding to the low dimensional space Y, we calculate the Mahalanobis distance $M_d$ as:

$$M_d(y_i) = (y_i - \mu)^T \Sigma^{-1} (y_i - \mu) \qquad (6)$$

Here, $\mu$ and $\Sigma$ are the mean and covariance matrix of the components in Y defined as true-positives. Components with large Mahalanobis distances are indicative of non-neuronal structures. Manually discarding these components allows refining the estimations of $\mu$ and $\Sigma$ and provides a more accurate sorting of components. This function is integrated in CaliAli with an intuitive user interface detailed in CaliAli's documentation.

### Algorithm details: enhancement of BV structures

BV structures were enhanced from raw one-photon Ca[+2] imaging frames using Hessian-based enhancement filters. In Ca[+2] imaging experiments, BVs typically appear as elongated structures with lower intensity values than the surrounding tissue. Hessian-based enhancement filters exploit these characteristics by examining the eigenvalues of the Hessian matrices of an image. The Hessian matrix has two eigenvalues at each pixel location. The relationship between these eigenvalues helps identify different structures in the image. In the case of BVs, the primary eigenvalue (λ1) is generally much smaller in magnitude than the secondary eigenvalue (λ2), indicating a tubular structure. In practice, the filtering is implemented as follows:

Step 1: Empirically determine the range of BV diameters $d_1$, $d_2$, ... $d_n$ in the raw image.

Step 2: For each diameter, perform steps 3 to step 6.

Step 3: Sequentially convolute the columns and rows of the image using a 1-D Gaussian filter with $\sigma = \frac{d_1}{4}, \frac{d_2}{4}, \ldots, \frac{d_n}{4}$.

Step 4: Compute the Hessian matrix of the filtered image:

$$H_2(x,y) = \begin{bmatrix} \frac{\partial^2 f(x,y)}{\partial x^2} & \frac{\partial^2 f(x,y)}{\partial x \partial y} \\ \frac{\partial^2 f(x,y)}{\partial y \partial x} & \frac{\partial^2 f(x,y)}{\partial y^2} \end{bmatrix} \quad (7)$$

Here, $f(x,y)$ is the intensity function of the filtered image at pixel $(x,y)$ and $\frac{\partial^2 f(x,y)}{\partial x^2}, \frac{\partial^2 f(x,y)}{\partial x \partial y}$, etc. are second-order partial derivatives of $f$. For faster computation, we calculated the Hessian matrix using the implementation described by Yang et al. [41]

Step 5: Find the eigenvalues of the Hessian matrix. The eigenvalues of $H_2(x,y)$ are obtained from the following analytical equation:

$$\lambda_1 = \frac{1}{2}\left(-\alpha_1 + \sqrt{\alpha_1^2 - 4\alpha_2}\right), \lambda_2 = \frac{1}{2}\left(-\alpha_1 - \sqrt{\alpha_1^2 - 4\alpha_2}\right) \quad (8)$$

Here, $\alpha_1$ and $\alpha_2$ are the roots of the characteristic polynomial of $H_2(x,y)$ given by:

$$\alpha_1 = -\left(\frac{\partial^2 f(x,y)}{\partial x^2} + \frac{\partial^2 f(x,y)}{\partial y^2}\right), \alpha_2 = -\left(\frac{\partial^2 f(x,y)}{\partial x^2}\frac{\partial^2 f(x,y)}{\partial y^2} - \frac{\partial^2 f(x,y)}{\partial x \partial y}\frac{\partial^2 f(x,y)}{\partial y \partial x}\right) \quad (9)$$

Step 6: Calculate the filter response function, defined as the larger absolute value between $\lambda_1$ and $\lambda_2$:

$$\phi(x,y) = \begin{cases} \lambda_1, & \text{if } |\lambda_1| > |\lambda_2| \\ \lambda_2, & \text{if } |\lambda_1| \le |\lambda_2| \end{cases} \quad (10)$$

Step 7: To obtain the enhanced vasculature image, we applied the filter response function to the original image by combining the results from multiple scales (i.e., different vessel diameters or filter sizes):

$$BV_{enhanced} = \sum_{i=1}^{n} \sigma_i \phi(x,y) \quad (11)$$

We considered 10 diameter sizes ranging from 2.4×gSiz to 3.5×gSiz, where gSiz is the filter size (in pixels) defined in CNMF-E.

## Algorithm details: log-demons registration

Sessions were registered using the diffeomorphic log-demons algorithm[15], which is an image registration method that computes a smooth and invertible deformation field to match a moving image to a fixed reference image. The algorithm is based on the original demons algorithm by Thirion[14], which was extended to ensure diffeomorphism (i.e., smoothness of the deformation). The optimal displacement field $S$ that aligns a moving image M to the static image $F$ is estimated by minimizing the velocity field $\vec{v}$ in the following energy function:

$$E(\vec{v}) = ||F - M \circ (S + \exp(\vec{v}))||^2 + \frac{\sigma_i^2}{\sigma_x^2} || \exp(\vec{v})||^2 \quad (12)$$

The velocity field $\vec{v}$ is used to additively update $S$ in each iteration, and $\sigma_i^2$ and $\sigma_x^2$ are weights regulating the similarity term $||F - M \circ (S + \exp(\vec{v}))||^2$ and the maximum step in each iteration, respectively. The algorithm uses an exponential map to update the deformation field to ensure that it remains diffeomorphic. Here, $\exp(\vec{v})$ is the exponential map of the vector field $\vec{v}$.

$\vec{v}$ is calculated using the additive demon algorithm:

$$\vec{v} = \frac{(F-M)\,\overline{\nabla} M}{(\overline{\nabla} M)^2 + (F-M)^2}.$$

Note that the numerator indicates that the moving image is displaced in proportion to its gradients ($\overline{\nabla} M$) and the alignment quality

$(F - M)$, which becomes 0 when the images perfectly match. The denominator normalizes the weighted gradient, ensuring that the magnitude of the deformation field is appropriate and not overly influenced by regions with extremely high intensity differences or gradients.

Two regularization steps are included at each iteration. For a fluid-like regularization, we convolute $\vec{v}$ with a Gaussian kernel with size = $\sigma_{fluid}$. For a diffusion-like regularization, we convolute $S$ with a Gaussian kernel with size = $\sigma_{diffusion}$. The regulation parameters used at each registration level are $\sigma_{fluid} = (1, 1, 3, 3)$, $\sigma_{diffusion} = (5, 5, 3, 3)$, $\sigma_x^2 = (1, 1, 1, 2)$, and $\sigma_i^2 = (1, 1, 1, 1)$.

## Reporting summary

Further information on research design is available in the Nature Portfolio Reporting Summary linked to this article.

## Data availability

The data supporting the plots in this study are provided in the Source Data file. Ca²⁺ imaging videos, simulation code, and additional source data that exceed the file size limit of the Source Data file are available on Dryad under the accession code: https://doi.org/10.5061/dryad.crjdfn3ck. Data sets produced in other studies can be obtained according to the data availability statements of the original manuscripts: Dual-color imaging data: https://doi.org/10.1101/2024.07.03.601770. Place cells data: https://doi.org/10.1016/j.crmeth.2022.100207. The multiplane imaging data used in this study is the property of Inscopix and was used with their permission. This dataset (SampleMultiplane_DS_TPC_data) is available for download to registered Inscopix users through their website: https://iqlearning.inscopix.com/software-downloads/sample-datasets.

## Code availability

The source code for CaliAli, along with demo videos and tutorials, is available on GitHub: https://github.com/CaliAli-PV/CaliAli. A preserved version of the code at the time of publication has been archived on Zenodo: https://doi.org/10.5281/zenodo.14934924. The code used to create video simulations is available on Dryad under the accession code: https://doi.org/10.5061/dryad.crjdfn3ck.

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

## Acknowledgements

This work was partially supported by the Japan Agency for Medical Research and Development (JP21zf0127005, JP23wm0525003), Japan Society for the Promotion of Science (JSPS)(24H00894, 23H02784, 22H00469, 16H06280, 20H03552, 21H05674, 21F21080), Takeda Science Foundation, Uehara Memorial Foundation, The Mitsubishi Foundation, and G-7 Scholarship Foundation to M.S., JSPS (23K19393, 24K18212) to I.K. and the Japan Science and Technology Agency (JPMJSP2124) to Y.W. We thank K.G. Akers for comments on the manuscript, S. Hasegawa, N. Tomita, and S. Hirase (Physiotech Inc.) for technical assistance, and M. Sakurai and I. Sekiguchi for secretarial support.

## Author contributions

Conceptualization, P.V., and M.S.; Methodology, P.V., Y.W., D.K, Y.C, Y.S, and M.S.; Investigation, Y.W., P.V., S.S., Z.D., Y.F., I.K., D.K., Y.C, T.N., T. Shuman, D.C., and M.S.; Validation, P.V., Y.W., and M.S.; Formal Analysis, P.V., and M.S.; Data Curation, P.V., Y.W., and M.S.; Visualization, P.V.; Writing – Original Draft, P.V.; Writing – Review & Editing, P.V., and M.S.; Funding Acquisition, M.S., M.Y., T. Sakurai, Y.S., P.V, Y.W., I.K. and M.K.; Resources, M.S., M.Y., T. Sakurai, and M.K.; Supervision, M.S., and M.Y.; Project Administration, M.S., P.V., and M.Y. All authors discussed and approved the manuscript.

## Competing interests

T-scope is produced by Physiotech Inc., which licenses the technology from the University of Tsukuba. One of the authors, M.S., receives licensing revenue from the University of Tsukuba. This author affirms that this financial interest has not influenced the research outcomes or interpretations presented in this study. The remaining authors declare no competing interests.
