## [Transparent Peer Review file · Nature Communications]

A Comprehensive Suite for Extracting Neuron Signals Across Multiple Sessions in One-Photon Calcium Imaging

Corresponding Author: Dr Masanori Sakaguchi

Version 0:

Reviewer comments:

Reviewer #1

(Remarks to the Author)

The authors develop a tool (CaliAli) for long-term tracking of neuronal population dynamics in calcium imaging data. This method differs from previous methods in two main aspects: 1) CaliAli performs a non-rigid alignment of imaging data across sessions based on blood vessels within the field of view in addition to the neuron shapes that were used in previous methods; 2) Unlike previous methods where neural signals are extracted from each session separately and only then aligned to one another, CaliAli extracts neural signals from concatenated videos of different sessions that were aligned prior to concatenation. The authors show that their method significantly outperforms previous methods based on different types of simulated data and based on an example of a real data set from the dentate gyrus. While the methodology is sound and the suggested approach was not previously used for registering cells across sessions in calcium imaging, current image registration (motion correction) methodologies for 1-photon calcium imaging are typically based on spatially high-pass filtered data (Giovannucci et al, 2019) in which the blood vessels within the field are enhanced, and the concatenation approach they use was also previously implemented as part of the batch mode used in the CNMF-e original paper (Pengcheng et al., 2018). Moreover, although the authors successfully show that CaliAli is superior to previous methods for the specific conditions they examine in the manuscript, I am concerned that their method might not perform as well under different conditions the authors do not acknowledge in their study although such conditions are frequently observed in calcium imaging data. If the authors appropriately test their methods under these other conditions mentioned in the comments below and revise their manuscript to clearly state in which conditions each method exhibits better performance, CaliAli could serve as a useful tool for the calcium imaging community that complements existing cell registration and cell identification methods.

Major comments:

1. The authors assume that calcium imaging data typically exhibit 15-25 μ m maximum non-rigid alignment. While the authors do show this in extended data fig. 1 based on real calcium imaging data, this amount of non-rigid alignment seems rather large for 1-photon calcium imaging. In this study, the authors only compare non-rigid alignment with translations. However, it may be possible that the existence of unmeasured rotations across sessions (which frequently occur in some versions of the miniaturized microscopes) could explain the large seemingly non-rigid shifts observed here. The authors should compare their non-rigid alignment with translations + rotations and if indeed rotations explain their previous observations of large non-rigid shifts, they should revise their manuscript (and most importantly their simulated data) accordingly. If rotations do not explain the large non-rigid shifts, it would still make their case for such large non-rigid shifts stronger if this comparison was included as part of extended data fig. 1.

2. Unlike previous cell registration methods, CaliAli extracts neural signals from concatenated videos that were aligned prior to concatenation. As the authors mention, such an approach is indeed advantageous in its ability to differentiate between inactive and undetected neurons. While the approach presented here does allow for the identification of calcium activity with low SNR that might otherwise go undetected, the authors don't discuss or at least mention the disadvantages of this approach:

I) The concatenation approach provides no indication of whether the data is suitable for tracking neurons across sessions. One of the biggest issues with longitudinal calcium imaging studies, especially those that record across days and weeks, is that the focal plane (z axis) can change across sessions. In such cases where z-plane changes significantly, no matter what approach or algorithm will be used to align the videos or register the cells across days, a certain degree of errors is inevitable. When the different sessions are concatenated and the spatial footprints are identified together for the entire concatenated video, there will be no indication of the sessions not resembling one another. This indication is on the other hand provided when identifying cells separately for each session and registering the identified cells across the different sessions, which allows validating the stability of the preparation across sessions. It would be important to see how that method performs under cases with significant changes in the focal plane across sessions.

II) If there are true changes to the cell's spatial footprint across sessions then an identification from a concatenated session would rely on the average of the cell's spatial footprint across the two sessions. Separated cell detection on the other hand does not average out true changes in the spatial footprints of cellular activity across sessions, and such a session-specific cell identification may lead to more accurate extraction of the cell's spatial footprint and temporal dynamics for each session. The authors can test their method under conditions of changing spatial footprints across sessions (independent of the FOV shifts).

III) The accuracy of the identification of cells (in any cell detection algorithm) is based on the contributions of the cell to the overall fluctuations in fluorescence within the FOV. When a session with a cell with low SNR is concatenated to a session with the same cell with high SNR, the identification accuracy of the concatenated session (while surely better than for the low SNR session) might be lower than that of the high SNR session. An analysis showing the differences between different approaches for cells with low activity sessions and high activity sessions would be interesting.

IV) Concatenating the calcium videos from multiple sessions into a single video is not scalable, as at some point the data size would become too big for most computers to handle and cell identification would become impossible. The combination of separated cell identification followed by cell registration makes this process scalable to extremely large data sets without reaching computational limits. It would be useful if the authors specify the total data set size (the concatenated video in terms of number of frames and number of x,y pixels per frame) that could be processed with a given computer (with specific RAM etc.) using their concatenation and batch-processing approach. These possible disadvantages should be addressed by the authors to provide a fairer depiction of the novelty and significance of their work in the context of previous work in the field.

3. CaliAli aligns imaging data across sessions based also on blood vessels and not only neuron shapes as were used in previous methods. The authors state that given the homogenous shapes of neurons, alignment in previous methods is more susceptible to local minima. Since blood vessels display higher structural heterogeneity, relying on them for alignment across sessions is more suitable for correcting larger displacements in the field of view. While this rationale is true, the authors don't acknowledge an issue that frequently hinders algorithms for within-session motion correction in calcium imaging which also rely on blood vessels. In some imaging data sets, there are no prominent blood vessels that can be used as fixed and heterogenous landmarks for image alignment. The authors should address this issue by demonstrating how CaliAli performs under such conditions.

4. The authors test their method on real imaging data consisting of 5-minute sessions, but not on longer imaging sessions. Moreover, their recordings are from the dentate gyrus where neuronal activity is temporally sparse. Together, these factors mean that many of the recorded cells are active only very few times throughout each session or even across all sessions. Under such conditions, it would be harder for cell identification algorithms such as CNMF, that rely on the spatiotemporal statistics of cellular activity within the recording, to accurately identify the cells in each session separately. Concatenating the sessions into a longer session alleviates this issue by providing more data based on which the cells can be identified. However, for imaging sessions much longer than 5 minutes (or from more active brain regions), in which the added value of concatenation becomes less significant, it may be that CaliAli does not outperform previous methods that rely on separate cell identification for each session. Thus, the authors should test their methods on longer imaging sessions as well to better assess its superiority under such conditions.

5. In Fig. 2n-q the authors show that CaliAli yields more cells that are active in all sessions and more correlated population vectors across sessions compared with previous methods. This result is not surprising given that CaliAli performs a unified cell identification for concatenated sessions, ensuring that the same cells are tracked across the two sessions (overlap in detected cells across sessions is 1 by definition), and thereby increasing correlation across sessions. However, as mentioned in the previous comments, concatenating sessions could result in less accurate cell identification due to the non-specificity of the extracted spatial footprint to each session. This could in turn result in lower accuracy/consistency of the signals of the cells within a given imaging session. Therefore, although cell detection of concatenated data increases the estimated coding stability across sessions, it may yield lower stability within a session compared to separated cell detection, indicating it could compromise detection accuracy. An analysis of the PV correlations within each session (e.g., first half of the session versus second half or some other temporal division of each session) would provide a more complete picture of the accuracy of CaliAli in tracking neurons over time.

6. The authors state that their method incorporates information from blood vessels and neurons to correct for inter session misalignment. In extended data fig 2 they further explain that they utilized a multilevel approach in which larger displacements are calculated on coarser versions of blood vessel images, and precise adjustments are calculated using neuron shapes. While this is a very important part of their methodology, the authors do not go into the details of how this multi-level approach is implemented, not in the main text nor in the materials and methods section. Thus, it is hard to

understand how for example neuron shapes are used for precise adjustments before cell identification (which requires that data to be already corrected for motion artifacts) is performed. The authors should better explain how they use both neuron shapes and blood vessels to align the different sessions and what is the exact sequence of processing steps required to reproduce their work.

Minor comments:

1. A legend explaining Fig. 2g is missing.
2. Statistics should be added to Fig. 1e-f and Fig. E4b.

Reviewer #2

(Remarks to the Author)

The authors provide a calcium imaging analysis tool focusing on the registration of population neurons across the long-term recording, which is an important scenario in modern neuroscience research. The method is very straightforward and seems useful.

Cell registration can be a very trivial problem if we have perfect alignments of the field of view (FOV), which is the task of motion correction in calcium imaging analysis. In practice, aligning all FOVs in multiple sessions is computationally intensive and may face unexpected issues if FOVs show large distortions. Thus tools like CellReg extract neurons first and then align these neurons based on the spatial footprints of all extracted neurons. I believe the authors worried about inaccurate FOV alignment from these extracted neurons while other features like blood vessels should be considered as well.

(1) The main drawback of this method is novelty. The abstract highlighted the main contribution as employing an alignment-before-extraction strategy incorporating vasculature information. This strategy is, in principle, the same as concatenating all sessions with good motion corrections and then extracting neurons. For example, a neuron paper "Anxiety Cells in a Hippocampal-Hypothalamic Circuit" used this strategy as well. As for the incorporation of vasculature information, this has been explicitly or implicitly used in most motion correction algorithms. Thus, I think the novelties of this method are very limited. Of course, the computation issue exists and should be taken care of. Actually, I think the writing of this manuscript focused on trivial points and it should be improved by highlighting the following contributions:

- A complete computational suite for processing long-term recording of calcium imaging data. It can output registered neurons and their activity in all sessions
- Significant efforts have been spent on improving quality, speed, and memory usage compared to cnmf-e.
- Align-before-registration yields better cell extraction and avoids false registrations caused by inaccurate registrations of cellreg.

The current version uses too many words in the abstract and the introduction to criticize current neuron-tracking algorithms. In my opinion, it's just an issue of FOV alignment and they can easily fix this issue by aligning FOVs with images of vasculatures.

(2) What if no blood vessel exists? This is a usual case in calcium imaging data.

Reviewer #3

(Remarks to the Author)

In this study, Vergara et al. present a new strategy for tracking neural activity using one-photon calcium imaging. Unlike prior research that relied solely on spatial footprints of regions of interest (ROIs) for neuron alignment, the authors propose a novel approach utilizing blood vessel (BV) patterns, which they argue yields improved alignment performance. They demonstrate that their method, CaliAli, surpasses the effectiveness of previous techniques in detecting identical neurons. Overall, this approach holds considerable promise in identifying neurons across multiple days. However, it is important to note that the current manuscript lacks sufficiently strong evidence to fully substantiate the authors' claims and my current conclusion is that this method is not more useful than previous ones for analysing in vivo (not simulated) data. Addressing the concerns outlined below would be crucial for a recommendation towards publication. Additionally, it is necessary to highlight that the data presentation and writing quality exhibit notable deficiencies. Substantial improvements are necessary for the manuscript to meet the expected standards.

Major

1. The authors should strive for a clearer explanation regarding the utilization of blood vessels to enhance registration performance, as this constitutes the central focus of this study. It may be advisable to incorporate a breakdown of Extended Data Fig 2 within the main results section. It remains unclear whether blood vessels are solely employed for aligning the overall field of views (FOVs). If this is indeed the case, and fine alignment primarily relies on neurons, the core argument of this study becomes trivial and also contradicts the assertion of an "alignment-before-extraction" method as stated in the abstract. If the authors intend to establish that their method has more than that, they must provide a more explicit elucidation of how this is accomplished. At present, the authors primarily assert the superior performance of their method compared to previous approaches without offering comprehensive details to support their claims.

2. If the correlation of BV patterns diminishes over consecutive days, as demonstrated in Fig 1j, and if neurons are registered based on BV patterns aligned across sequential sessions, it raises doubts about the accuracy of identifying neurons as the

same cells across distant time points (e.g., between day 1 and day 40). Because the spatial positioning of individual neurons in relation to a specific BV is likely preserved (in other words, the long-term displacement of BVs and neurons is not independent), different neurons are easily picked up if the alignment depends on BVs patterns especially with one-photon calcium imaging (see below). To become more positive for their arguments, it would be very important for the authors to experimentally verify their strategy. For example, they could employ sparse neuronal labeling techniques to assess whether registered pairs of neurons remain "anatomically" identical even after significant displacement of BVs. Otherwise, I regrettably remain far from being convinced in light of the provided information.

3. There is a notable lack of clarity as to why using blood vessel-aligned concatenated videos would result in improved signal-to-noise ratio (SNR), as indicated in lines 53-56. It is equally plausible that this approach captures either different neurons that subsequently overlap with the corresponding ROIs following the alignment of blood vessels between different sessions, or neuropil signals. This alternative possibility should not be overlooked, particularly considering the considerable axial point spread function inherent in one-photon imaging, which makes it challenging to exclude the influence of different neurons located outside the focal plane. This could potentially explain all the in vivo data presented in Fig 2. For instance, the optogenetic experiment could be interpreted in the following manner: CaliAli exhibits a higher consistency of neurons compared to other methods because it can consistently capture different neurons or neuropil, most of which respond to the optogenetic stimulus (Fig 2k). Essentially, it appears that there is a compromise in single-cell resolution with this method (although the authors seem to consider it as a higher SNR), which potentially accounts for the elevated population vector correlations observed in Fig 2q. As pointed out above, some experimental validation is necessary.

Minor

1. To ensure clarity, it is advisable to explicitly state, at the outset of the results sections, that you aim for registering cells after one-photon calcium imaging and the ROI segmentation relies on calcium activity (i.e. it can't be done by segmenting a static image like motion corrected average). Such information would be helpful for people unfamiliar with one-photon calcium imaging.

2. The data presentation and explanation in this work exhibit a significant lack of precision. The figures contain a multitude of unexplained information, as exemplified by the insufficient figure legend "d-f, Comparison of projections and their registration performance." This description alone falls far short of providing adequate clarification for the figure panels. While I haven't listed all the concerns herein, it is imperative that the authors enhance the data presentation for virtually all figures. Considering that this is the initial submission, it would be advantageous for the authors to provide more comprehensive details, without causing any detriment.

3. What is the ground truth data in Fig.1h and Fig. E5? If they are simulated, it is confusing to call them as the ground truth.

Version 1:

Reviewer comments:

Reviewer #1

(Remarks to the Author)

The authors develop a tool (CaliAli) for extracting neuronal population dynamics across multiple sessions in 1-photon calcium imaging data. As part of their revision, the authors now test CaliAli under a much wider range of conditions that are frequently observed in calcium imaging experiments, and more clearly highlight the novelty of their method and its contributions to other existing methods in the field. With these revisions, the manuscript was significantly improved, and the authors have addressed most of my concerns. However, I do have a few additional concerns. If the authors address these concerns, I believe CaliAli could serve as a useful tool for the calcium imaging community that complements existing cell extraction and cell registration methods.

Major comments:

1. In Supplementary Fig. 6e-f the authors find that CaliAli is more sensitive to detecting cases where the FOV is significantly changed across sessions. This higher sensitivity is demonstrated by the lower spatial correlations measured with CaliAli than with only footprints for a 62 μ m focal plane change (Supplementary Fig. 6e), and the higher PVD measured with CaliAli compared with CellReg or SCOUT for a 23 μ m focal plane change (Supplementary Fig. 6f). The authors interpret these results as an overestimation of stability when using only footprints or when using CellReg or SCOUT and that this overestimation can be prevented by using CaliAli. However, since there is no ground truth for this data set and the true stability is unknown, an alternative explanation for these results is that when there are focal plane changes across sessions, the footprints alignment or CellReg or SCOUT are more accurate in finding true similarities across sessions than when using BVs. The spatial correlations in Supplementary Fig. 6e are more convincing since they are higher for CaliAli than only footprints when focal plane changes are smaller (6 μ m or 23 μ m) but are lower for CaliAli than footprints when focal plane change is larger (62 μ m). The PVD in Supplementary Fig. 6f however, is inconsistent with this result since unlike spatial correlation for the smaller change in focal plane of 23 μ m the PVD is already higher for CaliAli than for CellReg or SCOUT. Since 1-photon calcium imaging exhibits very limited optical sectioning, it may be that for a 23 μ m focal plane change, some of the detected cells are still the same across sessions. Thus, since the results in Supplementary Fig. 6f are not consistent with the results in Supplementary Fig. 6e and since there is no ground truth, it is unknown if the higher stability observed with CellReg or SCOUT for a 23 μ m is an overestimation or a detection of true stability. Using the same rationale, the authors could have concluded for example in Fig. 8 that the lower PVDs observed with CaliAli compared with other methods are an

overestimation of stability. Therefore, my suggestion is that the authors find a more convincing way to show that this higher stability is indeed an overestimation (but not the case of overestimated stability in Figure 8). Alternatively, the authors can remove Supplementary Fig. 6f and use only Supplementary Fig. 6e to show the higher sensitivity of CaliAli for detecting large changes (62 μ m) in focal plane across sessions.

2. In Figure 4l it seems like the performance CNMF batch is almost at chance level according to the F1 score. Since CNMF batch mode is widely used by the calcium imaging community, it is important that the authors explain what it is about the specific properties of their simulated data that makes CNMF batch fail in such a way and explain under which conditions CNMF batch and CaliAli should yield comparable results.

3. In Figure 6b the authors compare spatial information content in CA1 place cells between sessions analyzed after concatenation or independently. One limitation of analysis is that the estimation of information content is positively biased due to limited sample sizes, and this bias is higher for cells with lower activity rates. The authors should either show that the activity rates are not different between conditions (concatenation and independent session analysis) or add a control where they randomly dilute the activity to obtain the same activity levels across conditions before estimating information or use a bias correction method to obtain a bias-free estimation of information content. The authors should also consider presenting this data in a linear scale instead of logarithmic scale to allow for an easier comparison with spatial information quantification in previous studies.

Minor comments:

1. Lines 55-57: This sentence is not clear. The issue of non-rigid misalignment is worsened by the neuron's homogenous blob-like shapes. But it is unclear what can easily converge to local minima. I am assuming the authors meant that alignment algorithms can converge into local minima, but this is not clear from the way the sentence is currently written.

2. Line 116: The same issue. Perhaps "using neuron shapes for finding the optimal across-session alignment is susceptible to local minima."

3. Supplementary Fig. 1b: "Percentage of non-rigid alignment in relation to the cumulative misalignment" should be changed to "Percentage of non-rigid misalignment..."

4. Lines 105-107: I think it should be a distance metric and not a similarity metric. Perhaps it can just be written as: "the algorithm minimizes the difference between images".

5. Lines 130-133: The local correlation image is mentioned here for the first time as a representative of neuron shapes. It should be explained (with a reference to its usage in the CNMF-e paper or in older papers) how this operation yields enhanced neuron shapes as the authors have done with the explanation of how the Hessian filter enhances blood vessels.

6. Supplementary Fig. 8b and Supplementary Fig. 11b – should the panel titles be "F1" "Sensitivity" and "Precision" instead of "F1" "Recall" and "Precision" to be consistent with Fig. 3a?

7. Lines 349-351: representational drift in general is when the neuronal representation of the same stimuli gradually changes over time. The scenario in which two orthogonal representations gradually change over time while preserving information content is a specific case of representational drift simulated by the authors. The wording should be changed accordingly.

8. Supplementary Fig. 11b – should the type of experiment be "low overlap no remapping" instead of "ideal" to be consistent with Fig. 5a?

9. Lines 365-366: This sentence implies that this is the first time in the study the authors use in vivo calcium imaging data. However, Fig. 6 was already based on in vivo calcium imaging data. The authors should rewrite this sentence accordingly.

(Remarks on code availability)

Reviewer #2

(Remarks to the Author)

The authors have effectively addressed my earlier concerns, and the updated version significantly surpasses the initial one. All contributions were well presented.

(Remarks on code availability)

Reviewer #3

(Remarks to the Author)

The revised manuscript has significantly improved in terms of data presentation and provides a clearer description of how CaliAli functions. I believe this will offer readers a much better understanding of the method's advantages and

disadvantages. My main concern with the previous version was whether the calcium signals in a CaliAli-defined ROI originate from truly identical neurons or different sources (i.e., either different cells or neuropil signals). The authors have included some additional experiments and analyses in the revised version; however, most of these additions are minor, and there is still no quantitative assessment addressing this issue. Please refer to my comments below for further details. I conclude that while this method performs well on simulated data, it does not appear to offer any improvement over previously reported methods for analysing in vivo imaging data. It is ultimately up to the journal editors to decide whether this information warrants publication.

Here are my major comments on the revised manuscript:

1. There seems to be some misunderstanding regarding my suggestion to employ sparse labelling. In Supplementary Figure S12, the authors conducted a very trivial test by manually selecting identical cells in two separate sessions and comparing different methods for registration. Clearly, any method would perform well when registering two sets of cells that are visibly present in both images. My concern was whether it is genuinely possible to correctly register neurons whose positions change over time along with blood vessels (BVs). This can be addressed not merely by selecting cells from two images, but by applying CaliAli to sparsely labelled neurons where the xyz coordinates of individual neurons are available from post-hoc histological experiments, for example.

2. Related to the first point, Supplementary Figure S12 exemplifies a spatial bias where co-registered neurons (highlighted by white circles) are predominantly on the left side of the image, while those on the right side appear very different, likely due to different tilt angles between the two sessions. These neurons cannot be registered together as they are different. CaliAli takes an incorrect approach by attempting to register such sessions using a common reference image, as explained in the rebuttal (lines 482 to 486): "This functionality operates locally within the FOV, allowing different parts of the FOV to use distinct sessions as reference points. For example, the upper portion of the FOV may use session 2 as a reference, while the lower portion aligns with session 3" If cells are segmented in this way, ROIs drawn in the upper portion would either be silent or reflect activities of different cells (or neuropil) during session 3. This likely explains why CaliAli detects more cells than other methods and thus reports better place decoding, simply because there are more cells in total.

3. Supplementary Figure S13 is also trivial. Trackability must naturally increase after alignment. The critical question is how many false positives CaliAli generates. Additionally, why is the trackability without alignment (Figure S13c) equivalent to that of CellReg and SCOUT (Figure 7d)? This suggests that the previously reported methods do nothing and merely pick up neuropil signals. I am uncertain if the authors are making fair comparisons. The same concern applies to Figures 8b and 8c.

4. Supplementary Figure S13 further supports my concern that CaliAli likely picks up neuropil signals. Comparing ROI shapes is meaningless; it merely shows that neuropil ROIs do not resemble cells, which is true by definition and does not rule out the possibility that CaliAli primarily detects neuropil signals.

(Remarks on code availability)

Version 2:

Reviewer comments:

Reviewer #1

(Remarks to the Author)

The authors have adequately addressed all my concerns. I have no additional comments on the manuscript.

(Remarks on code availability)

REVIEWER COMMENTS

Reviewer #1 (Remarks to the Author):

The authors develop a tool (CaliAli) for long-term tracking of neuronal population dynamics in
calcium imaging data. This method differs from previous methods in two main aspects: 1) CaliAli
performs a non-rigid alignment of imaging data across sessions based on blood vessels within
the field of view in addition to the neuron shapes that were used in previous methods; 2) Unlike
previous methods where neural signals are extracted from each session separately and only
then aligned to one another, CaliAli extracts neural signals from concatenated videos of different
sessions that were aligned prior to concatenation. The authors show that their method
significantly outperforms previous methods based on different types of simulated data and
based on an example of a real data set from the dentate gyrus. While the methodology is sound
and the suggested approach was not previously used for registering cells across sessions in
calcium imaging, current image registration (motion correction) methodologies for 1-photon
calcium imaging are typically based on spatially high-pass filtered data (Giovannucci et al, 2019)
in which the blood vessels within the field are enhanced, and the concatenation approach they
use was also previously implemented as part of the batch mode used in the CNMF-e original
paper (Pengcheng et al., 2018). Moreover, although the authors successfully show that CaliAli is
superior to previous methods for the specific conditions they examine in the manuscript, I am
concerned that their method might not perform as well under different conditions the authors do
not acknowledge in their study although such conditions are frequently observed in calcium
imaging data. If the authors appropriately test their methods under these other conditions
mentioned in the comments below and revise their manuscript to clearly state in which
conditions each method exhibits better performance, CaliAli could serve as a useful tool for the
calcium imaging community that complements existing cell registration and cell identification
methods.

We thank the reviewer for their insightful comments on our manuscript. We rigorously improved
the manuscript to emphasize the novelty and widespread applicability of CaliAli across various
conditions.

To better describe the novelty of CaliAli, we substantially extended descriptions of how CaliAli
aligns sessions (lines 103-227; Figs. 2, 3) and processes concatenated videos (lines 259-330;
Fig. 4), emphasizing its differences from other strategies (lines 431-488; Supplementary Table
1).

In terms of widespread applicability, we incorporated new data, including CA1 place cell data
(lines 355-363, Fig. 6), redesigned our previous experiments in the DG using population activity
and optogenetics (Figs. 7, 8; Supplementary Fig. 13, 15), and conducted new experiments
demonstrating CaliAli's utility even when there are changes in the FOV due to z-axis
displacement across sessions (Supplementary Figs. 6, 9). We also provide a detailed
description of the advantages and disadvantages of CaliAli in relation to other methods in the
Discussion section (lines 431-488; Supplementary Table 1).

Major comments:

1. The authors assume that calcium imaging data typically exhibit 15-25 μ m maximum non-rigid
alignment. While the authors do show this in Supplementary fig. 1 based on real calcium
imaging data, this amount of non-rigid alignment seems rather large for 1-photon calcium

imaging. In this study, the authors only compare non-rigid alignment with translations. However,
it may be possible that the existence of unmeasured rotations across sessions (which frequently
occur in some versions of the miniaturized microscopes) could explain the large seemingly non-
rigid shifts observed here. The authors should compare their non-rigid alignment with
translations + rotations and if indeed rotations explain their previous observations of large non-
rigid shifts, they should revise their manuscript (and most importantly their simulated data)
accordingly. If rotations do not explain the large non-rigid shifts, it would still make their case for
such large non-rigid shifts stronger if this comparison was included as part of Supplementary fig.
1.

We updated our manuscript to include a comparison of non-rigid alignment with both
translations and rotations (Supplementary Fig. 1). On closer inspection, using maximum non-
rigid displacement to determine the amplitude of non-rigid misalignment appeared to be
sensitive to outlier values, possibly leading to overestimation of non-rigid misalignments. This
became particularly evident when rotations were incorporated, causing excessive displacement
in some pixels.

To address this, we adopted a more robust method by calculating non-rigid misalignment
amplitudes based on the 95th percentile of pixel displacement. With this revised approach, our
estimation of non-rigid misalignment within a 4-day period is approximately 8 μm . To further
illustrate the challenges arising during long-term tracking, we provide new data showing that
non-rigid misalignment increases over time, reaching 15 μm within a 28-day period in some
cases. We support these results with both DG and CA1 data (Supplementary Fig. 1d).

We also ensured that the non-rigid misalignments used in simulations comparing CaliAli to other
methods closely resemble our empirically measured values (4-day period, $\sim 8 \mu\text{m}$; detailed in
Supplementary Fig. 11a).

In experiments comparing alignment performance across various projections (Fig. 2), we
simulated a non-rigid misalignment with an amplitude of 12.5 μm , reflecting the deformation
magnitudes observed across large inter-session gaps (Supplementary Fig. 1d).

2. Unlike previous cell registration methods, CaliAli extracts neural signals from concatenated
videos that were aligned prior to concatenation. As the authors mention, such an approach is
indeed advantageous in its ability to differentiate between inactive and undetected neurons.
While the approach presented here does allow for the identification of calcium activity with low
SNR that might otherwise go undetected, the authors don't discuss or at least mention the
disadvantages of this approach:

I) The concatenation approach provides no indication of whether the data is suitable for tracking
neurons across sessions. One of the biggest issues with longitudinal calcium imaging studies,
especially those that record across days and weeks, is that the focal plane (z axis) can change
across sessions. In such cases where z-plane changes significantly, no matter what approach
or algorithm will be used to align the videos or register the cells across days, a certain degree of
errors is inevitable. When the different sessions are concatenated and the spatial footprints are
identified together for the entire concatenated video, there will be no indication of the sessions
not resembling one another. This indication is on the other hand provided when identifying cells
separately for each session and registering the identified cells across the different sessions,
which allows validating the stability of the preparation across sessions. It would be important to

see how that method performs under cases with significant changes in the focal plane across
sessions.

We added new data to specifically address issues related to change in the FOV by evaluating
the performance of CaliAli and other methods in recordings in which we systematically changed
the focal plane across sessions. These new data is presented in two sections of the manuscript:

First, we implemented an approach in which the stability of the FOV was evaluated by
comparing the neuron projection (i.e., correlation images) corresponding to each session after
alignment. This approach is similar to the CellReg strategy that utilizes neuron footprints instead
of correlation images. Our findings suggest that by including BVs, we can avoid overestimating
neuron overlap and ensure that neurons from different focal planes are not incorrectly aligned
(lines 244-249; Supplementary Note 1; Supplementary Fig. 6). This allows CaliAli to detect FOV
changes to a similar or better extent than other methods.

Second, we corroborated that amid z-axis displacement, the neuron population vector distances
between sessions are proportional to changes in focal planes (i.e., degree of z-axis
displacement; Supplementary Fig. 6f). Finally, we confirmed that sessions recorded at different
focal planes do not impact the extraction of other sessions when videos are concatenated (lines
327-330; Supplementary Note 3; Supplementary Fig. 9). Overall, these results indicate that
CaliAli is resilient to error spread in the presence of large variations in the FOV.

II) If there are true changes to the cell's spatial footprint across sessions then an identification
from a concatenated session would rely on the average of the cell's spatial footprint across the
two sessions. Separated cell detection on the other hand does not average out true changes in
the spatial footprints of cellular activity across sessions, and such a session-specific cell
identification may lead to more accurate extraction of the cell's spatial footprint and temporal
dynamics for each session. The authors can test their method under conditions of changing
spatial footprints across sessions (independent of the FOV shifts).

To address this point, we simulated Ca^{2+} imaging videos in which neurons' spatial components
changed across sessions. The degree of these changes was based on actual data collected
over 32 days. Notably, we did not find changes in neuronal extraction performance due to
changes in neuronal footprints (lines 327-330; Supplementary Note 2; Supplementary Fig. 8).

III) The accuracy of the identification of cells (in any cell detection algorithm) is based on the
contributions of the cell to the overall fluctuations in fluorescence within the FOV. When a
session with a cell with low SNR is concatenated to a session with the same cell with high SNR,
the identification accuracy of the concatenated session (while surely better than for the low SNR
session) might be lower than that of the high SNR session. An analysis showing the differences
between different approaches for cells with low activity sessions and high activity sessions
would be interesting.

As suggested by the reviewer, we compared low- and high-SNR traces separately (lines 318-
326; Fig. 4n-p). We found a substantial increase in performance for low-SNR traces without
compromising the extraction of high-SNR traces.

However, as the reviewer suggests, processing concatenated video stacks may result in lower
extraction quality in some circumstances. CaliAli addresses this by improving the initialization
and matrix factorization steps of CNMF-E. Whereas standard CNMF-E batch implementations

follow a sequential batch processing strategy, CaliAli adopts a simultaneous batch processing
approach incorporating batch processing into each step of the NMF process. This substantially
improves the quality of extracted signals compared with the CNMF-E batch strategy (lines 259-
317; Fig. 4).

IV) Concatenating the calcium videos from multiple sessions into a single video is not scalable,
as at some point the data size would become too big for most computers to handle and cell
identification would become impossible. The combination of separated cell identification
followed by cell registration makes this process scalable to extremely large data sets without
reaching computational limits. It would be useful if the authors specify the total data set size (the
concatenated video in terms of number of frames and number of x,y pixels per frame) that could
be processed with a given computer (with specific RAM etc.) using their concatenation and
batch-processing approach.

These possible disadvantages should be addressed by the authors to provide a fairer depiction
of the novelty and significance of their work in the context of previous work in the field.

As suggested by the reviewer, we benchmarked CaliAli against CNMF-E in batch and standard
modes (lines 293-317; Fig. 4I). We note that in this manuscript, we fully implemented CaliAli in
batch mode, requiring only sufficient RAM to handle the largest imaging session and storing
final outputs (<2GB if the largest session was 180 × 260 pixels and 3,000 frames). This was
mainly achieved by redesigning the initialization and matrix factorization strategies in CNMF-E
as mentioned above (lines 293-317; Fig. 4). Additionally, we made minor adjustments to other
CaliAli modules (i.e., preprocessing, motion correction, inter-session alignment) to ensure
efficient memory management. Although these modifications are tribal and were not discussed
in the manuscript, they ensure that each step utilizes only the necessary memory to process
one session at a time.

3. CaliAli aligns imaging data across sessions based also on blood vessels and not only neuron
shapes as were used in previous methods. The authors state that given the homogenous
shapes of neurons, alignment in previous methods is more susceptible to local minima. Since
blood vessels display higher structural heterogeneity, relying on them for alignment across
sessions is more suitable for correcting larger displacements in the field of view. While this
rationale is true, the authors don't acknowledge an issue that frequently hinders algorithms for
within-session motion correction in calcium imaging which also rely on blood vessels. In some
imaging data sets, there are no prominent blood vessels that can be used as fixed and
heterogenous landmarks for image alignment. The authors should address this issue by
demonstrating how CaliAli performs under such conditions.

We created a new section addressing this specific issue (lines 228-243) and modified CaliAli to
still be useful in preparations in which BVs are not available. In its current form, CaliAli benefits
but does not exclusively depend on BVs for intersession alignment. Now, CaliAli automatically
detects when BVs are not useful for registration and switches to a neuron-only alignment mode.
This adjustment ensures alignment performance that is equal to or surpasses other methods.
We validated this approach through simulations in which BV visibility was gradually reduced
(Supplementary Fig. 5). Notably, even when BV visibility was reduced by 50%, we still observed
performance improvements compared with using only neurons. This underscores the beneficial
role of BVs, even in scenarios where they are less distinct.

4. The authors test their method on real imaging data consisting of 5-minute sessions, but not
on longer imaging sessions. Moreover, their recordings are from the dentate gyrus where
neuronal activity is temporally sparse. Together, these factors mean that many of the recorded
cells are active only very few times throughout each session or even across all sessions. Under
such conditions, it would be harder for cell identification algorithms such as CNMF, that rely on
the spatiotemporal statistics of cellular activity within the recording, to accurately identify the
cells in each session separately. Concatenating the sessions into a longer session alleviates
this issue by providing more data based on which the cells can be identified. However, for
imaging sessions much longer than 5 minutes (or from more active brain regions), in which the
added value of concatenation becomes less significant, it may be that CaliAli does not
outperform previous methods that rely on separate cell identification for each session. Thus, the
authors should test their methods on longer imaging sessions as well to better assess its
superiority under such conditions.

We designed a new experiment to directly measure the effect of session duration on CaliAli's
performance in recordings from the CA1 (lines 411-413; Supplementary Note 5; Supplementary
Fig. 14), a substantially more active region than the DG. First, we performed continuous Ca^{2+}
recording for 1 hour, establishing this recording as a pseudo-ground truth (pGT) dataset. Next,
we divided this recording into sessions lasting 5 or 15 min each. From these segmented
sessions, we extracted and tracked neuron signals across sessions using CaliAli or other
tracking methods. We then compared the reconstructed signals from these sessions with the
pGT. Importantly, signals obtained with CaliAli demonstrated greater similarity to the pGT
dataset compared with other methods, regardless of whether short or long imaging sessions
were employed. These findings highlight CaliAli's ability to enhance neuronal trackability across
short and long sessions, as well as in more active brain regions.

We also improved the population vector analysis in DG recordings by utilizing 10-min sessions
instead of 5-min sessions and extended the tracking duration to 99 days (lines 394-430; Fig. 8a-
c). In these refined conditions, we confirm that population vectors tracked with CaliAli exhibit
greater stability than those tracked with other methods. Additionally, we calculated population
vectors across sessions while focusing solely on neurons with detected activity across session
pairs. This approach aimed to minimize potential biases favoring CaliAli due to increased
detectability of neurons. CaliAli still attained higher population vector stability than other
methods. This suggests that enhancements in performance are not solely attributed to session
concatenation under conditions of sparse neuronal activity.

5. In Fig. 2n-q the authors show that CaliAli yields more cells that are active in all sessions and
more correlated population vectors across sessions compared with previous methods. This
result is not surprising given that CaliAli performs a unified cell identification for concatenated
sessions, ensuring that the same cells are tracked across the two sessions (overlap in detected
cells across sessions is 1 by definition), and thereby increasing correlation across sessions.
However, as mentioned in the previous comments, concatenating sessions could result in less
accurate cell identification due to the non-specificity of the extracted spatial footprint to each
session. This could in turn result in lower accuracy/consistency of the signals of the cells within
a given imaging session. Therefore, although cell detection of concatenated data increases the
estimated coding stability across sessions, it may yield lower stability within a session compared
to separated cell detection, indicating it could compromise detection accuracy. An analysis of

the PV correlations within each session (e.g., first half of the session versus second half or
some other temporal division of each session) would provide a more complete picture of the
accuracy of CaliAli in tracking neurons over time.

As suggested by the reviewer, we compared the first half of each session against the second
half in the new 10-min data. Our observations revealed that CaliAli consistently yielded higher
population vector correlations within sessions. However, interpreting these results is challenging
due to variations in the number and properties of neurons detected by CaliAli and other
methods across sessions. The concatenation approach may detect neurons missed by other
methods, potentially inflating correlation values. To prevent overestimating CaliAli performance,
we first calculated the average PVD across the same sessions and then subtracted these
values from the PVD calculated across different sessions. Under these conditions, we still
observed that CaliAli generates more stable long-term population vectors (lines 394-408; Fig.
8d, e). We further confirmed that this increased PV stability is not an artifact caused by
concatenation by calculating PVD in concatenated data in which inter-session misalignments
were not corrected (Fig. 8b), as this resulted in substantially larger PVD. These results indicate
that high population vector stability is only achieved when data are properly aligned.

To further corroborate that CaliAli does not hinder within-session stability, we analyzed CA1
place cell data from a previous study with four distinct recording sessions (lines 355-363; Fig.
6). Our results demonstrate that the spatial information content of neurons within individual
sessions remains consistent and, in some cases, may even be enhanced when sessions are
concatenated and processed together using CaliAli. We also found that CaliAli improves spatial
decoding within and across sessions. We believe these new data indicate that CaliAli does not
hinder the extraction quality in individual sessions, as the ability to code for place is not
accessible during the neuronal extraction process.

6. The authors state that their method incorporates information from blood vessels and neurons
to correct for inter session misalignment. In Supplementary fig 2 they further explain that they
utilized a multilevel approach in which larger displacements are calculated on coarser versions
of blood vessel images, and precise adjustments are calculated using neuron shapes. While this
is a very important part of their methodology, the authors do not go into the details of how this
multi-level approach is implemented, not in the main text nor in the materials and methods
section. Thus, it is hard to understand how for example neuron shapes are used for precise
adjustments before cell identification (which requires that data to be already corrected for
motion artifacts) is performed. The authors should better explain how they use both neuron
shapes and blood vessels to align the different sessions and what is the exact sequence of
processing steps required to reproduce their work.

We include a summary figure outlining the exact sequence that the CaliAli pipeline uses to
process multi-session imaging experiments (Fig. 1). We have also developed comprehensive
online documentation (available at <https://caliali-pv.github.io/CaliAli/>) outlining each step
necessary to extract concatenated calcium signals from raw Ca²⁺ imaging data. Moreover, we
added a dedicated section in the main body of the manuscript to highlight how CaliAli utilized
neurons and BVs in a multiscale approach to correct intersession misalignments (lines 126-170;
Fig. 2).

Minor comments:

1. A legend explaining Fig. 2g is missing.

We corrected this in the revised manuscript.

2. Statistics should be added to Fig. 1e-f and Fig. E4b.

We added statistical details to these figure legends in the revised manuscript.

Reviewer #2 (Remarks to the Author):

The authors provide a calcium imaging analysis tool focusing on the registration of population
neurons across the long-term recording, which is an important scenario in modern neuroscience
research. The method is very straightforward and seems useful.

Cell registration can be a very trivial problem if we have perfect alignments of the field of view
(FOV), which is the task of motion correction in calcium imaging analysis. In practice, aligning all
FOVs in multiple sessions is computationally intensive and may face unexpected issues if FOVs
show large distortions. Thus tools like CellReg extract neurons first and then align these
neurons based on the spatial footprints of all extracted neurons. I believe the authors worried
about inaccurate FOV alignment from these extracted neurons while other features like blood
vessels should be considered as well.

(1) The main drawback of this method is novelty. The abstract highlighted the main contribution
as employing an alignment-before-extraction strategy incorporating vasculature information.
This strategy is, in principle, the same as concatenating all sessions with good motion
corrections and then extracting neurons. For example, a neuron paper "Anxiety Cells in a
Hippocampal-Hypothalamic Circuit" used this strategy as well. As for the incorporation of
vasculature information, this has been explicitly or implicitly used in most motion correction
algorithms. Thus, I think the novelties of this method are very limited. Of course, the
computation issue exists and should be taken care of. Actually, I think the writing of this
manuscript focused on trivial points and it should be improved by highlighting the following
contributions:

- A complete computational suite for processing long-term recording of calcium imaging data. It
can output registered neurons and their activity in all sessions

- Significant efforts have been spent on improving quality, speed, and memory usage compared
to cnmf-e.

- Align-before-registration yields better cell extraction and avoids false registrations caused by
inaccurate registrations of cellreg.

The current version uses too many words in the abstract and the introduction to criticize current
neuron-tracking algorithms. In my opinion, it's just an issue of FOV alignment and they can
easily fix this issue by aligning FOVs with images of vasculatures.

We appreciate the reviewer's recommendations. We made substantial modifications to highlight
two principal contributions of CaliAli: (1) a robust inter-session alignment module for correcting

misalignments over extended periods and (2) a tailored pipeline for neuronal signal extraction
from concatenated sessions, both of which are integrated in a complete suite to process multi-
session Ca²⁺ imaging data. We now place particular emphasis on the novelty of CaliAli in
relation to existing methods (Supplementary Table 1).

Regarding the following comments:

- A complete computational suite for processing long-term recording of calcium imaging data. It
can output registered neurons and their activity in all sessions

Aside from a complete restructuring of the manuscript to convey this message, we included a
summary figure (Fig. 1) depicting all steps that CaliAli undertakes to process raw Ca²⁺ imaging
videos into long-term tracked Ca²⁺ sequences. We have also developed comprehensive online
documentation (available at <https://caliali-pv.github.io/CaliAli/>) outlining each step necessary to
extract concatenated calcium signals from raw Ca²⁺ imaging data.

- Significant efforts have been spent on improving quality, speed, and memory usage compared
to cnmf-e.

We extended the description of how CaliAli extract signals from concatenated videos in a
dedicated section (lines 259-330; Fig. 4). We clearly illustrate how CaliAli's approach is distinct
from CNMF-E and provide quantitative data demonstrating enhanced performance and reduced
memory demands.

- Align-before-registration yields better cell extraction and avoids false registrations caused by
inaccurate registrations of cellreg"

We extended the description of how CaliAli improves neuronal extraction through an align-
before-registration strategy (lines 259-330; Fig. 4). We also describe how CaliAli prevents the
false registration of neurons in greater detail (lines 126-170; Fig. 2). Furthermore, we expanded
the description of the weighted group-wise registration strategy utilized by CaliAli to align
multiple sessions, even when certain session pairs cannot be directly aligned (lines 197-227;
Fig. 3).

The effectiveness of these implementations is substantiated by several new experimental
datasets, including a redesign of our previous optogenetics experiments (lines 373-393; Fig. 7)
and long-term stability data from the DG (lines 394-430; Fig. 8). Additionally, we present new
place cell data (lines 355-363; Fig. 6) and evaluate CaliAli's performance in scenarios with
changes in the FOV due to z-axis displacement (lines 244-249; Supplementary Fig. 6 and lines
327-330; Supplementary Note 3; Supplementary Fig. 9).

Regarding "This strategy is, in principle, the same as concatenating all sessions with good
motion corrections and then extracting neurons. For example, a neuron paper "Anxiety Cells in
a Hippocampal-Hypothalamic Circuit" used this strategy as well", we acknowledge the
reviewer's point that video concatenation and subsequent extraction is an established approach
as demonstrated in several previous studies, including our own (Kumar et al., Neuron 107.3
(2020): 552-565). However, earlier implementations primarily relied on tools designed for single-
session motion correction and extraction, overlooking the unique challenges inherent in long-
term experiments. We included an explicit mention of these issues in the revised introduction
(lines 66-96). We emphasize that most tools suitable for correcting motion artifacts within

sessions are inadequate for addressing misalignments across sessions, particularly over long
periods of time. This is because aligning across sessions often requires correcting larger, non-
rigid misalignments in the FOV that may not resemble each other to the same extent as within a
single session. We now describe this in Supplementary Fig. 1d, where we show that non-rigid
inter-session misalignments increase over time. Similarly, tools that are suitable for extracting
neuronal signals in individual sessions prove inadequate and memory-inefficient when tasked
with extracting signals from concatenated video stacks. We describe how CaliAli addresses
these problems in new sections of the manuscript (inter-session alignment: lines 103-227, Fig.
2, 3; processing of concatenated videos: lines 259-330, Fig. 4).

Regarding “As for the incorporation of vasculature information, this has been explicitly or
implicitly used in most motion correction algorithms”, previous methods attempted to utilize BV
structures by applying high-pass filters. However, these techniques are not exclusive to BVs,
potentially enhancing other structures and negatively affecting inter-session alignment. This
issue is now explicitly mentioned in lines 110-125. Consistent with this, our results indicate a
marked decrease in performance when inter-session alignment relies on high-pass filters
compared with using Hessian filters integrated into CaliAli (lines 159-170; Fig. 2d-f).

(2) What if no blood vessels exist? This is a usual case in calcium imaging data.

We composed a new section addressing this specific issue (lines 228-243) and modified CaliAli
to still be useful in preparations in which BVs are not available. In its previous form, CaliAli
benefited but did not exclusively depend on BVs for intersession alignment. Now, CaliAli
automatically detects when BVs are not useful for registrations and switches to a neuron-only
alignment mode. This adjustment ensures alignment performance that is either equal to or
surpasses that of other methods. We validated this approach through simulations, gradually
reducing BV visibility (Supplementary Fig. 5). Notably, even when BV visibility is reduced by
50%, we still observed performance improvements compared with using only neurons. This
underscores the beneficial role of BVs, even in scenarios where they are less distinct.

Reviewer #3 (Remarks to the Author):

In this study, Vergara et al. present a new strategy for tracking neural activity using one-photon
calcium imaging. Unlike prior research that relied solely on spatial footprints of regions of
interest (ROIs) for neuron alignment, the authors propose a novel approach utilizing blood
vessel (BV) patterns, which they argue yields improved alignment performance. They
demonstrate that their method, CaliAli, surpasses the effectiveness of previous techniques in
detecting identical neurons. Overall, this approach holds considerable promise in identifying
neurons across multiple days. However, it is important to note that the current manuscript lacks
sufficiently strong evidence to fully substantiate the authors' claims and my current conclusion is
that this method is not more useful than previous ones for analysing in vivo (not simulated) data.
Addressing the concerns outlined below would be crucial for a recommendation towards
publication. Additionally, it is necessary to highlight that the data presentation and writing quality
exhibit notable deficiencies. Substantial improvements are necessary for the manuscript to meet
the expected standards.

We appreciate the reviewer's valuable feedback. We implemented substantial improvements in
the areas highlighted by the reviewer:

- • Evidence: We conducted new experiments to validate CaliAli's long-term tracking
capabilities by redesigning previous optogenetic (Fig. 7) and long-term tracking
experiments (Fig. 8) and incorporating new place cell data (Fig. 6).
- • Writing and data presentation: We significantly revised the figure legends and
description of the methodology and introduced new figures and sections in the main text.
These additions include detailed descriptions of how CaliAli corrects inter-session
misalignments (lines 103-227; Fig. 2, 3) and processes concatenated video sessions
using a computationally efficient approach (lines 259-330; Fig. 4). Additionally, we
included a new figure summarizing the entire CaliAli pipeline (Fig. 1).

These modifications are described in detail below.

Major

1. The authors should strive for a clearer explanation regarding the utilization of blood vessels to
enhance registration performance, as this constitutes the central focus of this study. It may be
advisable to incorporate a breakdown of Supplementary Fig 2 within the main results section. It
remains unclear whether blood vessels are solely employed for aligning the overall field of views
(FOVs). If this is indeed the case, and fine alignment primarily relies on neurons, the core
argument of this study becomes trivial and also contradicts the assertion of an "alignment-
before-extraction" method as stated in the abstract. If the authors intend to establish that their
method has more than that, they must provide a more explicit elucidation of how this is
accomplished. At present, the authors primarily assert the superior performance of their method
compared to previous approaches without offering comprehensive details to support their
claims.

We now clearly detail in the main results section how CaliAli employs a multiscale approach
integrating both blood vessels (BVs) and neurons (lines 126-170, Fig. 2; Supplementary Video
1). We wish to emphasize that CaliAli systematically utilizes images of BVs and neurons
together in a multiscale approach to correct non-rigid deformations across FOVs. Utilizing BVs
or neurons by themselves is in fact not sufficient to achieve the level of performance reached by
CaliAli (Fig. 2d-f).

The "alignment-before-extraction" method described in the manuscript involves utilizing images
of BVs obtained through Hessian filters and neuron projections obtained from maximum
projections and correlation images of each session. These images do not necessitate the
extraction of neuronal signals from videos and are employed to correct inter-session
misalignments. Once these misalignments are corrected, all sessions are concatenated, and
Ca^{2+} signals are extracted using a strategy optimized for processing long concatenated video
sequences (lines 259-330, Fig. 4). This strategy is opposite to the typical approach of extracting
neuronal signals from each session and then utilizing aligned neuronal footprints to track
neurons ("extraction-before-alignment"). To achieve effective extraction of Ca^{2+} signals from
concatenated video sequences, CaliAli presents a complete suite optimized for this process,
which is now illustrated in Fig. 1.

2. If the correlation of BV patterns diminishes over consecutive days, as demonstrated in Fig 1j,
and if neurons are registered based on BV patterns aligned across sequential sessions, it raises
doubts about the accuracy of identifying neurons as the same cells across distant time points
(e.g., between day 1 and day 40). Because the spatial positioning of individual neurons in

relation to a specific BV is likely preserved (in other words, the long-term displacement of BVs
and neurons is not independent), different neurons are easily picked up if the alignment
depends on BVs patterns especially with one-photon calcium imaging (see below). To become
more positive for their arguments, it would be very important for the authors to experimentally
verify their strategy. For example, they could employ sparse neuronal labeling techniques to
assess whether registered pairs of neurons remain "anatomically" identical even after significant
displacement of BVs. Otherwise, I regrettably remain far from being convinced in light of the
provided information.

As suggested by the reviewer, we evaluated CaliAli's inter-session registration capability by
manually identifying neuron pairs that were sparsely distributed and had distinguishable
anatomical features across sessions separated by more than 40 days (lines 365-372;
Supplementary Fig. 12). We evaluated registration performance by monitoring how the
centroids of visually identified neurons changed before and after registration with CaliAli or other
methods. We found that CaliAli aligned manually annotated neurons with comparable
performance to other methods. However, manual data annotation tended to favor neural pairs
that were sparsely distributed, possessed clear anatomical shapes, and did not remap across
sessions. This experimental setup may not fully represent actual challenges encountered in
long-term tracking, such as high overlap and neural remapping across sessions. Considering
this, we enhanced our previous optogenetic-tagging experiments to experimentally validate
CaliAli tracking performance. Specifically, we now assess the consistency of optogenetic
responses over several weeks (Fig. 7). We acknowledge that the reviewer has raised several
concerns regarding this experimental approach (i.e., comment 3). In response, we made
significant improvements to address these issues, as detailed below (i.e., response to comment
3).

We understand that the reviewer's primary concern is whether a BV-dependent alignment
approach would remain effective over extended time scales given the consistent decrease in BV
correlation over time. In response to this concern, we emphasize that the spatial correlation of
aligned BVs does not linearly relate to registration accuracy. In other words, even if BV
correlation decreases over time, inter-session registration is not linearly affected. We made
several modifications to illustrate this notion (lines 171-196; Fig. 3). Briefly:

- 1) We introduced a new metric called the "BV similarity score" to assess the suitability of
BVs for inter-session alignment (lines 171-196; detailed in the Materials and Methods
section). In our earlier submission, we used BV spatial correlation instead of the "BV
similarity score." We made this change because the absolute correlation value is
dependent on the specific experimental setup and cannot be generalized across
different recording conditions. For instance, changing the spatial down-sampling of
videos can affect the BV spatial correlation.
- 2) We assessed tracking performance using video simulations where the degree of BV
similarity was systematically reduced (Fig. 3d). Our data reveal a non-linear relationship
between tracking performance and BV similarity score. Specifically, we found that
tracking performance remains uncompromised when the BV similarity score is above
2.7. This implies that even if the BV similarity score decreases over time, as long as it
remains above 2.7, accurate tracking using BVs is achievable.

- 3) In actual Ca^{2+} imaging experiments (Fig. 3a), the BV similarity score remains above the
threshold of 2.7 for up to 99 days. This suggests that BVs can be reliably utilized for
accurate inter-session registration over an extended time period.
- 4) Previously, we calculated the BV similarity score using a fixed focal plane across
sessions. Now, we expanded our analysis to include measurement of the BV similarity
score over 99 days by dynamically adjusting the focal plane. This markedly reduced the
decay in BV similarity score across sessions. We present both sets of results to
demonstrate the potential outcomes obtained using microendoscopes with or without
focal adjustment.

Lastly, we highlight that CaliAli incorporates a unique strategy for aligning multiple (i.e., more
than two) sessions. CaliAli employs a weighted group-wise registration approach (lines 197-227;
Fig. 3e-i) that allows for accurate alignment even when certain session pairs cannot be directly
matched (e.g., if their BV similarity score is below 2.7). In essence, if sessions 1 and 3 cannot
be directly aligned but can both be aligned to session 2, CaliAli automatically aligns them to
session 2, ensuring precise tracking.

This functionality operates locally within the FOV, allowing different parts of the FOV to use
distinct sessions as reference points. For example, the upper portion of the FOV may use
session 2 as a reference, while the lower portion aligns with session 3. This approach ensures
accurate alignment across various regions of interest by prioritizing the most stable reference
session (lines 223-227; Fig. 3g-i).

3. There is a notable lack of clarity as to why using blood vessel-aligned concatenated videos
would result in improved signal-to-noise ratio (SNR), as indicated in lines 53-56. It is equally
plausible that this approach captures either different neurons that subsequently overlap with the
corresponding ROIs following the alignment of blood vessels between different sessions, or
neuropil signals. This alternative possibility should not be overlooked, particularly considering
the considerable axial point spread function inherent in one-photon imaging, which makes it
challenging to exclude the influence of different neurons located outside the focal plane. This
could potentially explain all the in vivo data presented in Fig 2. For instance, the optogenetic
experiment could be interpreted in the following manner: CaliAli exhibits a higher consistency of
neurons compared to other methods because it can consistently capture different neurons or
neuropil, most of which respond to the optogenetic stimulus (Fig 2k). Essentially, it appears that
there is a compromise in single-cell resolution with this method (although the authors seem to
consider it as a higher SNR), which potentially accounts for the elevated population vector
correlations observed in Fig 2q. As pointed out above, some experimental validation is
necessary.

To validate that CaliAli results do not occur as result of poor single-cell resolution, we made the
following improvements (lines 373-430; Fig. 7, 8; Supplementary Note 4; Supplementary Fig.
13, 15):

- 1) We demonstrated that the percentage of opto-responding neurons showing consistent
opto-activation across sessions (i.e., opto-consistency) is significantly lower in
concatenated videos where inter-session misalignments were not corrected compared
with properly aligned sessions. This suggests that inadvertent alignment of neurons

cannot account for the high opto-consistency achieved with CaliAli, as high opto-
consistency is observed only when videos are correctly aligned. Additionally, only a
minority of extracted components responded to light stimulation (Fig. 7e), making the
random alignment of opto-responding neurons unlikely. We also validated these
observations with new data by repeating the opto-tagging experiment across 4 weeks
(Fig. 7f, g).

2) The reviewer indicates that the high opto-consistency observed with CaliAli may be
attributed to a susceptibility to capture neuropil signals, which could exhibit strong
coupling to light stimulation. To illustrate the reviewer's point, we deliberately extracted
neuropile signals by initializing neurons in visually neuron-absent regions (Extended Fig.
13e; forced neuropile extraction). As predicted by the reviewer, this resulted in high opto-
consistency values (Extended Fig. 13e). However, these extracted neuropile signals can
be easily distinguished, as they display heterogeneous and elongated shapes that differ
from putative blob-like somatic signals. To quantify these structural differences, we
introduced a new metric that ranks neurons based on their spatial congruence,
distinguishing consistent somatic shapes over varied non-somatic shapes (i.e., neuropile
signals) (lines 331-338; Supplementary Fig. 10). We then applied this metric to compare
components extracted by: (1) independent extraction of the neural signals in each
session (control situation where concatenation-derived issues would not be present), (2)
extraction from concatenated sessions with CaliAli, and (3) deliberate extraction of
neuropile signals. The morphologies of the components extracted using CaliAli were not
significantly different from those obtained by processing each session independently.
However, they were noticeably distinct from those derived from forced neuropile
initialization (Supplementary Note 4; Supplementary Fig. 13f). These findings confirm
that the high opto-consistency of CaliAli is not merely a byproduct of neuropile signal
extraction. We believe these results may complement the idea that components
extracted by CaliAli remain "anatomically consistent," as indicated by the reviewer in the
previous comment.

**Regarding** "Essentially, it appears that there is a compromise in single-cell resolution with this
method (although the authors seem to consider it as a higher SNR), which potentially accounts
for the elevated population vector correlations observed in Fig 2q", as done with the opto-
tagging data, we showed that high population vector stability cannot be achieved in situations in
which single-cell resolution is compromised. To demonstrate this, we calculated population
vectors from concatenated videos in which inter-session misalignments were not corrected
(lines 400-403; Fig. 8a-c). This resulted in substantially lower population vector stability,
indicating that high population vector stability is only achieved when video sessions are properly
aligned and not by the inadvertent alignment of neurons.

Note that there is a minor difference in the way we currently present the population vector data;
we previously measured the population vector correlation but now measure population vector
distance (lower distance relates to higher population vector correlations as detailed in the
Material and Methods section). Population vector distance is a more precise metric when the
properties and number of tracked neurons differ between conditions (i.e., CaliAli vs. other
methods). We also extended the time for which we tracked population vectors to up to 99 days

and doubled the session time from 5 to 10 minutes. These adjustments were done in response
to another reviewer's request (Reviewer 1, comment 5) and serve to avoid overestimating
CaliAli's performance given the different number of neurons obtained with different methods.

To further corroborate that CaliAli does not compromise single-cell resolution, we tested CaliAli
using CA1 place cell data (lines 355-363; Fig. 6). We used place cell data because the spatial
information encoded by place cells is independent of the Ca^{2+} extraction process. We found that
CaliAli does not affect the spatial information content of individual neurons within individual
sessions. Moreover, decoding of mouse position within and across session is improved when
neurons are tracked with CaliAli. Overall, these results support the notion that CaliAli does not
compromise single-cell resolution.

Regarding "There is a notable lack of clarity as to why using blood vessel-aligned concatenated
videos would result in improved signal-to-noise ratio (SNR), as indicated in lines 53-56", we
agree that our previous manuscript failed to adequately emphasize two key contributions of
CaliAli: a robust inter-session alignment module for correcting inter-session misalignments and
a tailored extraction pipeline for neuronal signal extraction from concatenated sessions. The
enhanced SNR of transients is not due to the inclusion of BVs but rather the result of processing
concatenated video stacks. We address this issue in a dedicated section of the manuscript
(lines 259-330; Fig. 4). Essentially, processing concatenated videos allows for the extraction of
low-SNR signals by identifying the spatial components of neurons during periods of high activity.
This shared spatial information across sessions facilitates the extraction of low-SNR Ca^{2+}
signals that would otherwise remain undetected if neuronal extraction were conducted
independently for each session.

Minor

1. To ensure clarity, it is advisable to explicitly state, at the outset of the results sections, that
you aim for registering cells after one-photon calcium imaging and the ROI segmentation relies
on calcium activity (i.e. it can't be done by segmenting a static image like motion corrected
average). Such information would be helpful for people unfamiliar with one-photon calcium
imaging.

We include this statement in lines 260-263.

2. The data presentation and explanation in this work exhibit a significant lack of precision. The
figures contain a multitude of unexplained information, as exemplified by the insufficient figure
legend "d-f, Comparison of projections and their registration performance." This description
alone falls far short of providing adequate clarification for the figure panels. While I haven't listed
all the concerns herein, it is imperative that the authors enhance the data presentation for
virtually all figures. Considering that this is the initial submission, it would be advantageous for
the authors to provide more comprehensive details, without causing any detriment.

To address the issue, we rewrote all figure legends and included comprehensive details in the
Material and Methods section.

3. What is the ground truth data in Fig.1h and Fig. E5? If they are simulated, it is confusing to
call them as the ground truth.

In the previous manuscript, we used the term “ground truth” in our simulations to maintain
consistency with the terminology used in previous studies that evaluated tracking performance,
such as the works of Sheintuch et al. 2017 (CellReg) and Johnston et al. 2022 (SCOUT). To
underscore that the term “ground truth” in our manuscript specifically pertains to a benchmark
established based on simulated data and may not necessarily represent an ultimate measure of
performance when applied to real data, we now utilize the phrase “simulated ground truth”
instead of “ground truth” when appropriate throughout the manuscript.

REVIEWER COMMENTS

Reviewer #1 (Remarks to the Author):

The authors develop a tool (CaliAli) for extracting neuronal population dynamics across multiple sessions in 1-photon calcium imaging data. As part of their revision, the authors now test CaliAli under a much wider range of conditions that are frequently observed in calcium imaging experiments, and more clearly highlight the novelty of their method and its contributions to other existing methods in the field. With these revisions, the manuscript was significantly improved, and the authors have addressed most of my concerns. However, I do have a few additional concerns. If the authors address these concerns, I believe CaliAli could serve as a useful tool for the calcium imaging community that complements existing cell extraction and cell registration methods.

Major comments:

1. In Supplementary Fig. 6e-f the authors find that CaliAli is more sensitive to detecting cases where the FOV is significantly changed across sessions. This higher sensitivity is demonstrated by the lower spatial correlations measured with CaliAli than with only footprints for a 62 μ m focal plane change (Supplementary Fig. 6e), and the higher PVD measured with CaliAli compared with CellReg or SCOUT for a 23 μ m focal plane change (Supplementary Fig. 6f). The authors interpret these results as an overestimation of stability when using only footprints or when using CellReg or SCOUT and that this overestimation can be prevented by using CaliAli. However, since there is no ground truth for this data set and the true stability is unknown, an alternative explanation for these results is that when there are focal plane changes across sessions, the footprints alignment or CellReg or SCOUT are more accurate in finding true similarities across sessions than when using BVs. The spatial correlations in Supplementary Fig. 6e are more convincing since they are higher for CaliAli than only footprints when focal plane changes are smaller (6 μ m or 23 μ m) but are lower for CaliAli than footprints when focal plane change is larger (62 μ m). The PVD in Supplementary Fig. 6f however, is inconsistent with this result since unlike spatial correlation for the smaller change in focal plane of 23 μ m the PVD is already higher for CaliAli than for CellReg or SCOUT. Since 1-photon calcium imaging exhibits very limited optical sectioning, it may be that for a 23 μ m focal plane change, some of the detected cells are still the same across sessions. Thus, since the results in Supplementary Fig. 6f are not consistent with the results in Supplementary Fig. 6e and since there is no ground truth, it is unknown if the higher stability observed with CellReg or SCOUT for a 23 μ m is an overestimation or a detection of true stability. Using the same rationale, the authors could have concluded for example in Fig. 8 that the lower PVDs observed with CaliAli compared with other methods are an overestimation of stability.

Therefore, my suggestion is that the authors find a more convincing way to show that this higher stability is indeed an overestimation (but not the case of overestimated stability in Figure 8). Alternatively, the authors can remove Supplementary Fig. 6f and use only Supplementary Fig. 6e to show the higher sensitivity of CaliAli for detecting large changes (62 μ m) in focal plane across sessions.

We acknowledge that the main issue with **Supplementary Fig. 6f** is the lack of ground truth to validate our claims. To address this, we added two new panels demonstrating that CellReg and SCOUT introduce unrealistic displacement in the x-y axis to match neurons across different focal planes (current **Supplementary Fig. 6f, g, lines 158-165 in Supplementary Note 1**).

Our rationale is as follows: In the experiment shown in **Supplementary Fig. 6**, we acquired imaging sessions consecutively without repositioning the microscope. This minimized potential misalignment between sessions, making them comparable to the minor shifts that occur within a single session (which are corrected with standard motion correction algorithms).

Importantly, we added new data demonstrating that x-y misalignments within a session are consistent across focal planes (current **Supplementary Fig. 6f**). This finding allows us to use the displacement values from motion correction as a realistic estimate of potential misalignment between sessions. If CaliAli or footprints alignment tools produce estimates that significantly exceed these values, this indicates these tools might be exaggerating true x-y inter-session misalignments. This would lead to an overestimation of neuronal overlap.

To confirm this, we compared the misalignment magnitudes estimated by CaliAli across focal planes with the within-session estimates and found no statistically significant difference. In contrast, footprints alignment methods overestimated the misalignment amplitudes (current **Supplementary Fig. 6g**). This further supports our assertion that other methods overestimate the overlap of neurons by introducing unrealistic x-y displacements.

We recognize that the PVD analysis does not directly relate to FOV alignment in the presence of z-axis displacement, as it lacks a straightforward interpretation. We believe the newly added data and analysis more effectively convey the figure's main point and have therefore removed the PVD analysis as recommended by the reviewer.

2. In Figure 4l it seems like the performance CNMF batch is almost at chance level according to the F1 score. Since CNMF batch mode is widely used by the calcium imaging community, it is important that the authors explain what it is about the specific properties

of their simulated data that makes CNMF batch fail in such a way and explain under which conditions CNMF batch and CaliAli should yield comparable results.

CNMF batch's lower performance stems from its reliance on repeated neuron detection from residual video data. This process, inherently more error-prone, becomes problematic when analyzing multiple sessions. Although negligible with few sessions, these errors accumulate, significantly impacting extraction performance when processing many sessions (e.g., 40 as in **Fig. 5I**). This is now explicitly addressed in **lines 323-331** and illustrated in a new supplementary figure:

"Subsequent batches use previously identified spatial components for matrix factorization, and new components are added from the residual, which is obtained by subtracting the neuronal signals of already identified components, leaving only undetected components (**Supplementary Fig. 8a**). However, the residual often includes non-somatic signals (i.e., dendrites) that could display a high spatial correlation and PNR and be incorrectly initialized as new neurons. These errors would further propagate as subsequent batches are processed (**Supplementary Fig. 8b-c**). Although negligible with few batches, this accumulation can significantly degrade performance with multiple batches (e.g., more than five) (**Supplementary Fig. 8c**)."

Furthermore, we clarify that the comparison between CaliAli and CNMF was conducted under conditions prone to high error propagation (i.e., utilizing 40 batches) (lines 340-344):

"To compare our approach with existing methods under conditions prone to high error propagation, we simulated 40 consecutive imaging sessions (5 min each). We then analyzed this concatenated data using two methods: CNMF with sequential batch processing (CNMF_{batch}) and standard CNMF processing of the entire video sequence at once (CNMF_{standard})."

3. In Figure 6b the authors compare spatial information content in CA1 place cells between sessions analyzed after concatenation or independently. One limitation of analysis is that the estimation of information content is positively biased due to limited sample sizes, and this bias is higher for cells with lower activity rates. The authors should either show that the activity rates are not different between conditions (concatenation and independent session analysis) or add a control where they randomly dilute the activity to obtain the same activity levels across conditions before estimating information or use a bias correction method to obtain a bias-free estimation of information content. The authors should also consider presenting this data in a linear scale instead of logarithmic scale to allow for an easier comparison with spatial information quantification in previous studies.

To address the concern regarding potential biases in the estimation of spatial information content due to differences in neuronal activity rates, we employed two complementary approaches:

1. Controlling for activity in the regression model: We included the average activity of each neuron as a covariate in our regression model (**Fig. 6b, lines 388-392**). This ensures that the calculated main effect of CaliAli versus independent session extraction is independent of any activity-related effects.

2. Standardizing information scores with a surrogate distribution: We standardized the information score values by generating a surrogate distribution for each neuron. This was done by randomly shuffling the activity traces within each session, creating a distribution of information scores representing chance-level performance given each neuron's specific activity level. The original information scores were then expressed as standard deviations above this chance level (**Fig. 6c**). This standardization effectively removes any bias related to differences in activity rates.

Using both approaches, we found no significant differences in the spatial information content of neurons extracted with CaliAli compared with those extracted from independent session analysis, indicating that the spatial information content is comparable between the two methods.

As requested by the reviewer, we now also present the spatial information content data on a linear scale.

To further strengthen our analysis, although not requested by the reviewer, we proactively addressed a previous observation that CaliAli yielded higher spatial information content than independent session analysis in only one session. Suspecting that this session-specific difference might be due to unaccounted variability, we expanded our analysis to include three mice (previously one) and incorporated "mouse" as a random effect in our linear mixed-effects model. With this improved analysis, we confirm that the spatial information content is comparable between sessions processed with CaliAli and those processed independently with CNMF-E. Furthermore, our updated analysis demonstrates that CaliAli maintains within-session decoding accuracy (**Fig. 6e**) while significantly enhancing across-session decoding (**Fig. 6k-l**). These improvements provide a more robust and comprehensive evaluation of CaliAli's performance, demonstrating its ability to enhance across-session analysis without compromising within-session data quality.

Minor comments:

1. Lines 55-57: This sentence is not clear. The issue of non-rigid misalignment is worsened by the neuron's homogenous blob-like shapes. But it is unclear what can easily converge to local minima. I am assuming the authors meant that alignment algorithms can converge into local minima, but this is not clear from the way the sentence is currently written.

We corrected this in the manuscript (**lines 57-59**).

2. Line 116: The same issue. Perhaps "using neuron shapes for finding the optimal across-session alignment is susceptible to local minima."

We corrected this in the manuscript (**lines 119-120**).

3. Supplementary Fig. 1b: "Percentage of non-rigid alignment in relation to the cumulative misalignment" should be changed to "Percentage of non-rigid misalignment..."

We corrected this in the figure legend.

4. Lines 105-107: I think it should be a distance metric and not a similarity metric. Perhaps it can just be written as: "the algorithm minimizes the difference between images".

We corrected this in the manuscript (**lines 108-110**).

5. Lines 130-133: The local correlation image is mentioned here for the first time as a representative of neuron shapes. It should be explained (with a reference to its usage in the CNMF-e paper or in older papers) how this operation yields enhanced neuron shapes as the authors have done with the explanation of how the Hessian filter enhances blood vessels.

We corrected this in the manuscript (**lines 138-144**).

6. Supplementary Fig. 8b and Supplementary Fig. 11b – should the panel titles be "F1" "Sensitivity" and "Precision" instead of "F1" "Recall" and "Precision" to be consistent with Fig. 3a?

We corrected this in the manuscript (**current Supplementary Fig. 9 and Fig. 12**).

7. Lines 349-351: representational drift in general is when the neuronal representation of the same stimuli gradually changes over time. The scenario in which two orthogonal representations gradually change over time while preserving information content is a specific case of representational drift simulated by the authors. The wording should be changed accordingly.

We corrected this in the manuscript (**lines 378-383**).

8. Supplementary Fig. 11b – should the type of experiment be “low overlap no remapping” instead of “ideal” to be consistent with Fig. 5a?

We corrected this in the manuscript (current **Supplementary Fig. 12b**).

9. Lines 365-366: This sentence implies that this is the first time in the study the authors use in vivo calcium imaging data. However, Fig. 6 was already based on in vivo calcium imaging data. The authors should rewrite this sentence accordingly.

We removed this sentence from the manuscript

Reviewer #2 (Remarks to the Author):

The authors have effectively addressed my earlier concerns, and the updated version significantly surpasses the initial one. All contributions were well presented.

We thank the reviewer for their positive comments.

Reviewer #3 (Remarks to the Author):

The revised manuscript has significantly improved in terms of data presentation and provides a clearer description of how CaliAli functions. I believe this will offer readers a much better understanding of the method's advantages and disadvantages. My main concern with the previous version was whether the calcium signals in a CaliAli-defined ROI originate from truly identical neurons or different sources (i.e., either different cells or neuropil signals). The authors have included some additional experiments and analyses in the revised version; however, most of these additions are minor, and there is still no quantitative assessment addressing this issue. Please refer to my comments below for further details. I conclude that while this method performs well on simulated data, it does not appear to offer any improvement over previously reported methods for analysing in vivo imaging data. It is ultimately up to the journal editors to decide whether this information warrants publication.

We thank the reviewer for their time and consideration of our manuscript. We believe our previous rebuttal letter may not have fully conveyed the extent of the modifications made to address the reviewers' comments. To ensure clarity, we included a table summarizing major modifications done during both the current and previous revisions that directly or indirectly address the concern of whether a CaliAli-defined ROI originates from truly identical neurons or different sources:

Experiment	Figure in current submission	Type of data
First submission		
Performance evaluation using video simulations	Supplementary Fig. 12	Simulations
Optogenetic-tagging of neurons (within 1 day)	Fig. 7	Real data
Long-term tracking over 28 days	Replaced with new data	Real data
Second submission		
Place cell decoding	Fig. 6	Real data
Optogenetic tagging for 4 weeks with sparse chrimson expression	Fig. 7	Real data
Reconstruction of a segmented 1h-long video	Supplementary Fig. 13	Real data
Long-term tracking for 99 days	Fig. 8	Real data
Current submission		
Comparison of footprints aligned by 13 algorithms	Fig. 2	Real data
Evaluation of neuronal trackability with dual-color imaging	Fig. 4	Real data
Improvements to place cell data	Fig. 6	Real data
Quantification of neuropile signals using machine learning	Supplementary Fig. 14	Real data

Here are my major comments on the revised manuscript:

1. There seems to be some misunderstanding regarding my suggestion to employ sparse labelling. In Supplementary Figure S12, the authors conducted a very trivial test by manually selecting identical cells in two separate sessions and comparing different methods for registration. Clearly, any method would perform well when registering two sets of cells that are visibly present in both images. My concern was whether it is genuinely possible to correctly register neurons whose positions change over time along with blood vessels (BVs). This can be addressed not merely by selecting cells from two images, but by applying CaliAli to sparsely labelled neurons where the xyz coordinates of individual neurons are available from post-hoc histological experiments, for example.

To address the reviewer's main concern—"Is it genuinely possible to correctly register neurons whose positions change over time alongside blood vessels (BVs)?"—we included two sets of experiments that quantitatively evaluate neuronal registration *in vivo*: (1) Correlation of aligned footprints, and (2) Dual-color imaging. These experiments demonstrate that CaliAli correctly registers neurons despite changes in the field of view, as described below.

Experiment 1 - Correlation of aligned footprints:

We compared neuronal projections from actual Ca²⁺ imaging data aligned by CaliAli against those obtained using alternative methodologies (**Fig. 2d-f; lines 170-182**). To ensure a fair comparison, we utilized the same evaluation criteria employed in previous studies, including NoRMCorre (CalmAn) by Pnevmatikakis et al. (J. Neurosci. Methods 291 (2017): 83-94), MIN1PIPE by Lu et al. (Cell Rep. 23.12 (2018): 3673-84), and PatchWarp by Hattori

and Komiyama (Cell Rep. Methods 2.5 (2022)), among others. Specifically, we assessed registration by measuring the correlation of the aligned neuronal projections and the sharpness of the post-registration average image, referred to as " crispness."

We compared CaliAli with several commonly used algorithms for inter-session registration and motion correction:

- Translation (Most packages)
- Demon (CellReg, SCOUT)
- Patch phase correlation (Suit2p)
- Cross-correlation (CalmAn/Suite2p/noRMCorre)
- Log-Demon (MIN1PIPE)
- Patch affine (PatchWarp)

We also examined whether these algorithms perform better when we incorporate blood vessel (BV) information or neuronal projections. However, none of these modifications matched CaliAli's performance. This result indicates that CaliAli's advantage arises not merely from including BVs or neurons, but from the synergy of combining both within its framework.

Our results demonstrate that CaliAli outperformed all these algorithms based on established criteria applicable to real Ca^{2+} imaging data.

Experiment 2 - Dual-color imaging:

We applied CaliAli to data from a dual-color microendoscope capable of imaging neurons expressing both GCaMP and the stable nuclear marker tdTomato (**Fig. 4; lines 256-269**). A member of the UCLA Miniscope team recently demonstrated that using the static tdTomato signal is a reliable method for neuronal cross-registration in one-photon imaging (Dong, Z. et al, Biorxiv (2024); <https://doi.org/10.1101/2024.07.03.601770>). Unlike GCaMP signals that fluctuate with neuronal activity, tdTomato emits a stable red fluorescence, allowing for continuous visualization of neuronal somata even when cells are inactive. Using these data, we accurately matched tdTomato-expressing neurons to establish a near ground truth anatomical identification across sessions. We then compared CaliAli's performance to that of 12 other methodologies in aligning neuronal footprints using only the GCaMP signals. CaliAli produced significantly better alignment than all other methods as determined by the true neuron coordinates defined with the tdTomato signals.

Therefore, our dual-color imaging results confirm that CaliAli correctly registers neurons whose positions change over time along with BVs, demonstrating significantly higher matching accuracy compared with widely used algorithms.

In this revision, the reviewer suggests to evaluate trackability through post-hoc histological identification of neurons. Although this approach is commonly employed in two-photon imaging experiments with head-fixed animals, we identified several challenges that render it unfeasible for one-photon imaging data:

- **Lack of optical sectioning:** One-photon imaging does not allow for optical sectioning, making impossible to completely match neurons detected in Ca^{2+} imaging with those obtained in histology.
- **Barrel distortion:** Imaging using a GRIN lens introduces barrel distortion, which affects the shape of neurons and complicates matching based on anatomical features.
- **Timing of post-hoc identification:** Post-hoc histological matching is typically performed with data acquired shortly before sacrificing the animal. If post-hoc identification was used to evaluate neuronal tracking, this would necessitate matching images and histology across multiple weeks. In such cases, the same challenges that hinder neuronal tracking across sessions (i.e., changes in neuronal activity and increased non-rigid misalignment) would also impact imaging-histology matching.
- **Lack of established protocols:** Validated protocols for post-hoc histological identification of neurons in one-photon Ca^{2+} imaging are very limited. To our knowledge, only one study has attempted to match miniscope-detected neurons to histology (Anner, P., et al. *J. Neurosci. Methods* 341 (2020): 108765). In this study, BVs were used to align imaged neurons with histological data, employing the same filters as CaliAli. However, this approach limits our ability to evaluate the trackability of BVs, as they are inherently used as reference points to align the imaging data with the histological data. Additionally, their matching performance was around 60%, below the tracking performance of most algorithms, and we expect even lower performance for sessions acquired several weeks earlier.

For these reasons, we believe that the dual-color imaging experiment better addresses the reviewer's concern, as it allows us to anatomically identify neurons in each session rather than at a single time point after histological preparation. These two experiments (i.e., correlation of aligned footprints and dual-color imaging) have been added in replacement of the manual annotation performed previously.

2. Related to the first point, Supplementary Figure S12 exemplifies a spatial bias where co-registered neurons (highlighted by white circles) are predominantly on the left side of the image, while those on the right side appear very different, likely due to different tilt angles between the two sessions. These neurons cannot be registered together as they are different. CaliAli takes an incorrect approach by attempting to register such sessions using a common reference image, as explained in the rebuttal (lines 482 to 486): "This functionality operates locally within the FOV, allowing different parts of the FOV to use distinct sessions as reference points. For example, the upper portion of the FOV may use session 2 as a reference, while the lower portion aligns with session 3" If cells are segmented in this way, ROIs drawn in the upper portion would either be silent or reflect activities of different cells (or neuropil) during session 3. This likely explains why CaliAli detects more cells than other methods and thus reports better place decoding, simply because there are more cells in total.

The manual annotation data from former Supplementary Fig. S12 was replaced with the new data described above (now shown in **Fig. 2d-f and Fig. 4**). These new data are free from biases associated with manual annotation, allowing for the objective evaluation of methods using standardized metrics.

However, we understand the reviewer's concern that the observed dissimilarities in neuronal projections raise questions about CaliAli's ability to accurately demix neuronal signals, given its registration approach.

We have the following observations to support that this is indeed not the case:

- 1) **Variation in neuronal projections do not necessarily reflect instability in the FOV:** During one-photon imaging, particularly in regions such as the DG, where neurons are densely packed and activity is sparse, variations in neuronal activity are common and do not necessarily indicate focal plane instability. This is evidenced with the new data shown in **Fig. 4b**, where we detected significantly more stable neuronal projections obtained from tdTomato signals than GCaMP signals imaged in the same FOV. Note that to ensure FOV stability throughout our imaging sessions, we employed criteria comparable to, or exceeding, those used by packages such as CellReg. The specific methods used to assess and compare FOV stability are detailed in **Supplementary Fig. 6 and Supplementary Note 1**. Additionally, to provide visual confirmation of FOV stability, we included average frames from each imaging session overlaid with neuron and BV projections (**Supplementary Fig. 3a**). These projections correspond to the new data shown in **Fig. 2d-f**. While neuronal projections may exhibit some variation across sessions (**Supplementary Fig. 3b**), likely reflecting changes in neuronal activity, the consistent appearance of blood

vessels and the average frame strongly suggests a stable field of view (**Supplementary Fig. 3a**).

- 2) **CaliAli does not align sessions to a common reference image**, as do other methods like CellReg. We apologize if this misunderstanding was caused by our previous explanation. CaliAli aligns sessions to a coordinate system that maximizes similarity across all sessions. This system is not fixed to a single image but is derived by weighting structural information from the most stable regions across all sessions. Even if a neuron is not detected in one session, CaliAli can accurately predict its location using spatial information from neighboring neurons and BVs in other sessions. Although this method does not correct for neurons "disappearing" due to shifts in the z-axis (nor do other methods), it ensures accurate calculation of x-y displacements—an issue that previous methodologies do not address effectively. This improved x-y alignment allows CaliAli to more accurately detect changes in the FOV, as detailed in **Supplementary Fig. 6 and Supplementary Note 1**. The quantitative data supporting better x-y alignment in real Ca²⁺ imaging is provided by the new data described above (**Fig. 2d-f** and **Fig. 4**).

- 3) **CaliAli can correctly demix signals of different neurons across sessions**: In one-photon imaging, signals from neurons up to 50 μm apart can be captured simultaneously (Zhang, Y., et al. Nat. Methods 20.5 (2023): 747-754). Therefore, demixing signals from multiple planes should be addressed not only when tracking neurons but also during the processing of individual sessions. Modern Ca²⁺ imaging pipelines, including CaliAli, utilize a constrained non-negative matrix factorization (CNMF) algorithm to demix signals, even when neurons occupy overlapping positions in the x-y plane. We demonstrate that CaliAli accurately demixes signals with two independent experiments:
 - a. **Place cell data**: Successful prediction of an animal's position in subsequent sessions using a decoder trained on a single session demonstrates that CaliAli does not mix signals from different neurons or neuropil, as such mixing would impair place decoding by mixing place fields. Regarding the reviewer's suggestion that CaliAli's potentially higher cell count explains its improved place decoding, we clarified that CaliAli detects a similar number of neurons in this particular dataset (**Fig. 6d**). In this case, CaliAli's strength lies in its ability to track more neurons consistently across sessions (**Fig. 6g**) while preserving place field stability (**Fig. 6h**) but without changing the number of detections per session. To further address this concern, we

performed intra-session place decoding under three conditions: using all data, using only consistently tracked neurons, and downsampling to 100 neurons per method. CaliAli consistently showed superior intra-session decoding performance in all scenarios (**Fig. 6k-m**). The improvement in decodability demonstrates that CaliAli accurately identifies individual neurons and does not mix signals.

- b. Opto-tagging data:** Higher consistency in optogenetic responses across sessions indicates that CaliAli does not mix signals from different neurons. We are aware that the reviewer is concerned with neuropil contamination in this dataset. We address this concern in the reviewer's comment #4.

Collectively, based on the reviewer's feedback, we replaced the data in former Supplementary Fig. S12 with new data presented in **Fig. 2d-f and 4**. We also included additional analyses and figures (**Fig. 6d-m, Supplementary Fig. 3, Supplementary Fig. 6, and Supplementary Note 1**) to further support the effectiveness of CaliAli's registration and signal demixing capabilities.

3. Supplementary Figure S13 is also trivial. Trackability must naturally increase after alignment. The critical question is how many false-positives CaliAli generates. Additionally, why is the trackability without alignment (Figure S13c) equivalent to that of CellReg and SCOUT (Figure 7d)? This suggests that the previously reported methods do nothing and merely pick up neuropil signals. I am uncertain if the authors are making fair comparisons. The same concern applies to Figures 8b and 8c.

We made several modifications to address these concerns, including clarifications, new analyses, and refinements to our methodology, which are detailed below.

1. Addressing the core concern—false-positives in CaliAli: We understand that the reviewer's primary concern is the potential for false-positives generated by CaliAli. We think this point is addressed by several *in vivo* experiments included in the current and previous submissions:

- a. Place cell data (Fig. 6; lines 384-403):** Concatenating sessions with CaliAli yields cells with spatial properties comparable to those processed independently. This demonstrates that CaliAli does not increase false-positives while improving intra-session place decoding, indicating enhanced across-session neuronal extraction. Importantly, place cell analysis is the standard method used by previous methodologies like CellReg and SCOUT to evaluate trackability.

- b. **1-Hour video reconstruction (Supplementary Fig. 13; lines 404-413):** We performed an interrupted 1-hour-long video recording. We then segmented this video into 5- or 10-min-long segments and tested the performance of different tracking strategies to reconstruct the original signals extracted from the 1-hour-long video. Although only in short time scales, this method provides direct estimation of false-positives (and false-negatives) in terms of F1 scores, which demonstrate that CaliAli improves trackability in short time scales.
 - c. **Evaluation of neuronal projections aligned with multiples algorithms (Fig. 2d-f; lines 170-182):** We utilized established metrics to evaluate the quality of aligned neuronal footprints with CaliAli versus 12 different methodologies, ensuring accurate neuronal alignment and minimizing false-positives.
 - d. **Dual-color imaging (Fig. 4; lines 257-269):** This experiment quantifies alignment performance by comparing GCaMP-based neuron alignment to the ground truth established by stable tdTomato expression.
 - e. **Opto-tagging experiments (Fig. 7; 414-444):** False-positives in these experiments are quantified as the proportion of neurons with inconsistent optogenetic responses.
2. **Clarification of "without alignment":** We acknowledge that our use of the phrase 'without alignment' was ambiguous. Previously, this phrase meant deactivating CaliAli's non-rigid alignment module, while still applying basic translations. To address this ambiguity, **Fig. 7e (line 425-428)** now shows CaliAli's performance when we systematically introduced increasing levels of misalignment to the field of view (FOV). This demonstrates that true "chance level" performance occurs when misalignment is substantial (i.e., above 15 μm). These data are now shown next to CellReg and SCOUT data (**Fig.7d**) to clearly demonstrate that these methods are not performing at chance level. For other panels in **Fig. 7 and 8**, "misaligned data" now consistently refers to data with intentionally introduced large shifts (15 μm) in the FOV, simulating a condition approaching "chance level" performance.
3. **Purpose of former Supplementary Fig. 13c and its relation to opto-consistency:** Former Supplementary Fig. 13c (now replaced with **Fig. 7e**) originally aimed to address a concern raised in the first revision: "CaliAli exhibits a higher consistency of neurons compared to other methods because it can consistently capture different neurons or neuropil, most of which respond to the optogenetic stimulus". While we acknowledge that trackability naturally increases after alignment, the reviewer's initial comment indicated a skepticism regarding opto-consistency as a valid measure of trackability. To demonstrate the direct relationship between opto-consistency and accurate

alignment, we measured opto-consistency in *misaligned* data. This analysis reveals that opto-consistency is significantly reduced under misalignment conditions (current **Fig. 7e**). Therefore, higher opto-consistency cannot be explained by consistently capturing different neurons or neuropil; rather, it is a direct consequence of accurate neuronal alignment.

- 4. Ensuring fair comparison across methods:** To ensure fair and accurate comparison of CaliAli, CellReg, and SCOUT, we refined our analysis to address potential biases arising from differences in neuron detection. Specifically, we redefined our criteria for identifying "opto-inconsistent" neurons. A neuron is now considered opto-inconsistent only if a neighboring neuron within a 16- μm radius (i.e., average neuron size) exhibits a light response in the same session. This ensures that any observed inconsistencies in optogenetic responses are truly due to misalignment, and not simply an artifact of low neuron detection rates. Such low detection rates could disproportionately affect methods like CellReg and SCOUT, which tend to identify fewer neurons per session in conditions of sparse neuronal activity.

Interestingly, this modification significantly improved the measured opto-consistency with SCOUT but not with CellReg (**Fig. 7h**). We attribute this difference to the distinct strategies for alignment employed by these methods. As detailed in the manuscript (**lines 440-444**), the opto-tagging experiments utilized relatively sparse GCaMP expression to avoid neuronal crosstalk. Under such sparse conditions, registration algorithms like CellReg, which rely solely on spatial information from neuronal signals, often perform poorly (as demonstrated in **Supplementary Fig. 3e, f** with simulated data using neurons sparsely distributed), as they have fewer spatial cues to match across sessions. By contrast, SCOUT incorporates temporal information in addition to spatial features, making it less susceptible to the challenges of sparse data. CaliAli incorporates BVs, which are not affected by neuronal sparsity. In current **Fig. 8** (population vector analysis), where GCaMP expression is dense, both CellReg and SCOUT perform significantly better than the misaligned data but still do not reach the accuracy achieved by CaliAli.

- 5. Improvements to data presentation for population vector analysis:** In the previous version of **Fig. 8**, we illustrated two approaches to calculating population vectors: 1) the conventional method of discarding neurons with undetected activity in one or more sessions or 2) an alternative method in which undetected activity is treated as zero. Both approaches are necessary for population vector analysis due to its requirement for equal dimensionality across sessions, a constraint not inherently met by methods like CellReg and SCOUT, as they do not track all neurons across all sessions. Naturally,

assigning zero to undetected activity would increase the sparsity of population vectors calculated for CellReg and SCOUT, which would result in an overall larger population vector distance compared with CaliAli. To ensure that CaliAli is compared with other methods under the fairest conditions, we only present results utilizing the first method in **Fig. 8b, c**. The results associated with the second method are now presented in **Supplementary Fig. 15a, b**. We still include the second method in the manuscript as this approach is used in some studies and could be relevant to some readers.

4. Supplementary Figure S13 further supports my concern that CaliAli likely picks up neuropil signals. Comparing ROI shapes is meaningless; it merely shows that neuropil ROIs do not resemble cells, which is true by definition and does not rule out the possibility that CaliAli primarily detects neuropil signals.

We understand the reviewer's concern regarding the use of ROI shape as a metric in **Supplementary Fig. 13g** (current **Supplementary Fig. 14c**) and the potential for neuropil contamination. We agree with the reviewer that, by definition, neuropil ROIs would not resemble the shapes of cell bodies. The original figure included three datasets: ROIs from CNMF-E, ROIs from CaliAli, and intentionally extracted neuropil ROIs (current **Supplementary Fig. 14a**). The inclusion of the neuropil ROIs is intended as a negative control, demonstrating that our shape analysis could distinguish between neuronal and non-neuronal structures. However, the key comparison, which may have been inadvertently obscured, is between CaliAli and CNMF-E. In this specific context, the comparison of ROI shapes serves to demonstrate that CaliAli, when applied to concatenated sessions, extracts ROIs that are statistically indistinguishable from those extracted by CNMF-E from individual sessions. CNMF-E is a widely accepted standard for processing single-session one-photon data. Therefore, the similarity in ROI shapes between CaliAli and CNMF-E indicates that CaliAli is not introducing systematic bias toward non-neuronal shapes. Importantly, to be completely transparent, we present all ROIs extracted by each method in this analysis, not a selected subset (**Supplementary Fig. 14 f, and g**).

To further address the concern about neuropil contamination, we employed a machine learning algorithm trained to distinguish between neuronal and neuropil signals (**line 430-434, Supplementary Note 4**). This algorithm was trained on a dataset of manually labeled granule cells from the DG (imaged without optogenetic manipulation and processed with CNMF-E; **Supplementary Fig. 14d, e**), achieving accuracy comparable to existing tools used for neuropil identification in one-photon Ca^{2+} imaging (Tran et al., 2020).

Applying this algorithm to our opto-tagging data, we found no significant difference in the proportion of signals classified as "neuropil" between CaliAli and CNMF-E (**Supplementary Fig. 14f-h**). Again, to ensure complete transparency, we show all components extracted by the algorithm labeled as either "neuron" or "neuropil" in these figures. This quantitative analysis, using an unbiased machine-learning approach, directly demonstrates that CaliAli does not inflate the number of neuropil signals detected.

Taken together, these results—the similarity in ROI shapes to the established CNMF-E method and the unbiased machine-learning classification, all while presenting the complete set of extracted ROIs—strongly indicate that our optogenetic findings are not influenced by neuropil contamination. This further validates our opto-tagging approach as a reliable method for evaluating neuronal trackability.